# Exploiting Structure in Offline Multi-Agent RL: The Benefits of Low Interaction Rank

**Wenhao Zhan**[*]  **Scott Fujimoto**[†]  **Zheqing Zhu**[‡]

**Jason D. Lee**[§]  **Daniel R. Jiang**[†]  **Yonathan Efroni**[†]

## Abstract

We study the problem of learning an approximate equilibrium in the offline multi-agent reinforcement learning (MARL) setting. We introduce a structural assumption—the *interaction rank*—and establish that functions with low interaction rank are significantly more robust to distribution shift compared to general ones. Leveraging this observation, we demonstrate that utilizing function classes with low interaction rank, when combined with regularization and no-regret learning, admits *decentralized, computationally and statistically efficient* learning in offline cooperative and competitive MARL. Our theoretical results are complemented by experiments that showcase the potential of critic architectures with low interaction rank in offline MARL, contrasting with commonly used single-agent value decomposition architectures.

## 1 Introduction

Multi-agent reinforcement learning (MARL) is a general framework for interactive decision-making with multiple agents. Recent breakthroughs in this field include learning superhuman strategies in games like Go (Silver et al., 2016), StarCraft II (Vinyals et al., 2019), Texas hold'em poker (Brown and Sandholm, 2019), and Diplomacy (Bakhtin et al., 2022). Additionally, MARL has been successfully applied in real-world domains, including auctions (Jin et al., 2018), pricing systems (Nanduri and Das, 2007), and traffic control (Wu et al., 2017). However, most of these successes rely on online and iterative interaction with the environment, which enables the collection of diverse and exploratory data. In practice, online interaction with exploratory policies is often infeasible or prohibitive due to safety constraints, making it necessary to use offline datasets instead.

Several recent works have investigated the application of modern deep RL algorithms to the offline MARL setting (Yang et al., 2021; Tseng et al., 2022; Wang et al., 2024). Despite recent advances, there remains a lack of standardized methods that can effectively tackle complex, real-world problems beyond simulated or simplistic settings. Recent works (Cui and Du, 2022a; Zhang et al., 2023b) studied offline MARL from a sample complexity perspective. Specifically, Zhang et al. (2023b) designed the `BCEL` algorithm, a sample-efficient algorithm for the offline general-sum MARL setting with general function classes. However, its implementation poses significant challenges due to the need to solve a non-convex problem in the joint action space. Furthermore, the algorithm's sample complexity is tied to the unilateral coverage coefficient, which can scale exponentially with the number of agents in the worst-case scenario. This raises the following question, which becomes the focus of this work:

*Are there any natural structural assumptions that allow for both sample efficient and computationally efficient algorithms in the offline MARL setting?*

Recent lower bounds show that computing an equilibrium in a MARL setting is hard in general (Daskalakis et al., 2009; 2023). Nevertheless, for some specialized MARL classes, this need not

---

[*]Princeton University. Work done at Meta.

[†]Meta.

[‡]Pokee AI. Work done at Meta.

[§]Princeton University.

| Offline Setting | Reward Assumption | Sample Complexity | Efficient Algorithm |
|---|---|---|---|
| Markov Game | — | $O\left(C^N\right)$ | ✗ |
| Contextual Game | $K$-Interaction Rank | $O\left(C^K\right)$ | ✓ |
| Markov Game w/ Decoupled Transition | $K$-Interaction Rank | $O\left(C^K\right)$ | ✓ |

Table 1: Comparison of the results presented in this work (highlighted in orange) and prior work. $C$ is the single-agent coverage coefficient. Here we present the worst-case dependence of the sample complexity in the single-agent coverage coefficient, where $N$ is the number of agents.

be the case. In this work, we study the MARL setting with low *interaction rank* (IR). In this setting the reward model decomposes to a sum of terms, each involving the interactions of only a subset of the agents (Section 3). Our key statistical result is that functions with low interaction rank are more robust to distribution shift compared to general functions. This result, which, as we show, has natural applications in offline MARL, may also be of general interest.

Assuming the reward model has low interaction rank, we leverage regularization and no-regret learning to develop decentralized computationally-efficient offline algorithms for the contextual game (CG) setting, and for Markov games (MG) with a decoupled transition model (Section 4 and Section 5). Notably, we prove that applying structures with low interaction rank allows these algorithms to achieve sample-efficient learning, avoiding the exponential dependence in the number of agents. Lastly, in Section 6, we empirically corroborate our findings. This shows the potential of using reward architectures with low interaction rank in offline MARL setting, and the need to go beyond the standard single agent value decomposition architectures, which have been popularized for MARL (Sunehag et al., 2017; Rashid et al., 2020; Yu et al., 2022).

**Additional related works.** Our work is related to the research on provable offline and decentralized MGs and factored critics in the empirical literature. See Appendix A for more details.

## 2 PRELIMINARIES

We define the general offline multi-agent RL setting, which includes all of the models we study.

**General-sum contextual MG.** A contextual MG is defined by the tuple $\mathcal{M} = (N, H, \mathcal{C}, \mathcal{S} := \prod_{i=1}^{N} \mathcal{S}_i, \mathcal{A} := \prod_{i=1}^{N} \mathcal{A}_i, \{R_{i,h}^\star\}_{i=1,h=1}^{N,H})$ where $N$ is the number of agents and $H$ is the horizon. $\mathcal{C}$ is the context space. In each episode, a public context $c \in \mathcal{C}$, which is observed by all agents and stays invariant throughout the episode, is drawn from the distribution $\rho$. $\mathcal{S}_i$ and $\mathcal{A}_i$ are the local state and action spaces of the $i$-th agent. We assume the initial local state of each agent is fixed for simplicity, but our analysis can be easily extended to accommodate stochastic initial states. $R_{i,h}^\star(c, \boldsymbol{s}, \boldsymbol{a})$ is the reward distribution of agent $i$ at step $h$ given the context $c$, joint state $\boldsymbol{s}$ and joint action $\boldsymbol{a}$. We assume the value of $R_{i,h}^\star$ lies in $[0, 1]$ and denote the mean of $R_{i,h}^\star$ by $r_{i,h}^\star$. In this paper, we study *general-sum* RL (Littman, 1994) and thus $r_{i,h}^\star : \mathcal{C} \times \mathcal{S} \times \mathcal{A} \to [0, 1]$ can be an arbitrary reward function.

**MGs with decoupled transitions.** In this work we assume the transition of the local state only depends on the local state, public context and local action (Zhang et al., 2023a; DeWeese and Qu, 2024; Jin et al., 2024), which can be characterized by the kernel $P_{i,h}^\star : \mathcal{C} \times \mathcal{S}_i \times \mathcal{A}_i \mapsto \Delta_{\mathcal{S}_i}$ for all $i \in [N], h \in [H]$. Note that the reward function $R_{i,h}^\star(c, \boldsymbol{s}, \boldsymbol{a})$ is still of a general-sum game and depends on the joint state and joint action.

**Remark 1.** *The decoupled transitions property finds application in many practical scenarios including sensor coverage, autonomous vehicles, and robotics, and has been studied under online decentralized learning setting (Zhang et al., 2023a; DeWeese and Qu, 2024; Jin et al., 2024). For more general MGs, decentralized no-regret algorithms are hard to design even in full-information setting. As far as we know, Erez et al. (2023) is the only existing work which achieves sublinear regret in general MGs when all the agents adopt the decentralized algorithm. However, they only focus on tabular cases in the full information setting or online setting with a minimum reachability assumption. Therefore, we leave it as an important future direction to extend our analysis to more general MGs.*

**Policy and value functions.** A joint policy $\pi = \{\pi_h\}_{h=1}^H$ is a mapping from $\mathcal{C} \times \mathcal{S}$ to the simplex $\Delta_\mathcal{A}$ which determines the joint action selection probability under the public context and joint state at each step. Given $\pi$, we use $\pi_i = \{\pi_{i,h}\}_{h=1}^H$ to denote the marginalized policy for agent $i$. In the decentralized setting (Zhang et al., 2023a; DeWeese and Qu, 2024; Qu et al., 2020; Lin et al., 2021; Jin et al., 2024), each agent $i$ independently executes its *local policy* $\pi_i$ based only on the public context $c$ and its local state $s_i$, i.e., $\pi = \prod_{i=1}^N \pi_i$ where $\pi_{i,h} : \mathcal{C} \times \mathcal{S}_i \to \Delta_{\mathcal{A}_i}$ for all $i, h$. In this case, we call the joint policy $\pi$ a *product policy*.

Given a reward function $r_i$ of agent $i$ and joint policy $\pi$, we define the value function and Q-function associated with agent $i$ to be agent $i$'s expected return conditioned on the current joint state (and action):

$$V_{i,h}^{\pi,r}(c, \boldsymbol{s}) := \mathbb{E}_\pi \left[ \sum_{h'=h}^H r_{i,h'}(c, \boldsymbol{s}_{h'}, \boldsymbol{a}_{h'}) \,\bigg|\, c, \boldsymbol{s}_h = \boldsymbol{s} \right],$$

$$Q_{i,h}^{\pi,r}(c, \boldsymbol{s}, \boldsymbol{a}) := \mathbb{E}_\pi \left[ \sum_{h'=h}^H r_{i,h}(c, \boldsymbol{s}_{h'}, \boldsymbol{a}_{h'}) \,\bigg|\, c, \boldsymbol{s}_h = \boldsymbol{s}, \boldsymbol{a}_h = \boldsymbol{a} \right].$$

Here, $\mathbb{E}_\pi[\cdot]$ denotes the expectation under the distribution of the trajectory when executing $\pi$ in $\mathcal{M}$. We will omit the superscript $r$ in $V_{i,h}^{\pi,r}$ and $Q_{i,h}^{\pi,r}$ if $r$ is the ground truth reward $r^\star$.

**Decentralized offline equilibrium learning.** For any joint policy $\pi$, if each agent cannot increase its own expected reward by changing its policy while the other agents fix their policies, then $\pi$ is a coarse correlated equilibrium (CCE) (Aumann, 1987). More specifically, let $\Pi_i := \{\mu_i : \mathcal{C} \times \mathcal{S}_i \to \Delta_{\mathcal{A}_i}\}$ denote the local policy class of the agent $i$, then an $\epsilon$-approximate CCE can be defined as follows:

**Definition 1** (Coarse Correlated Equilibrium). *A joint policy $\pi$ is called an $\epsilon$-approximate CCE if*

$$\mathsf{Gap}_i(\pi) := \max_{\mu_i \in \Pi_i} \mathbb{E}_{c \sim \rho}[V_{i,1}^{\mu_i \times \pi_{-i}}(c, \boldsymbol{s}_1)] - \mathbb{E}_{c \sim \rho}[V_{i,1}^\pi(c, \boldsymbol{s}_1)] \leq \epsilon, \quad \forall i \in [N],$$

*where $\pi_{-i}$ is the marginalized policy of $\pi$ for all agents excluding $i$.*

If $\pi$ is a product policy and satisfies Definition 1, then $\pi$ is the well-known Nash equilibrium (NE) (Nash et al., 1950). Given the fact that NE can be hard to compute for even general-sum normal games (Daskalakis et al., 2009), our goal is to learn an $\epsilon$-approximate CCE.

In particular, we consider the *decentralized* setting where each agent $i$ executes its policy $\pi_i$ only based on the public context $c$ and its local state $s_i$ (Zhang et al., 2023a; DeWeese and Qu, 2024; Qu et al., 2020; Lin et al., 2021; Jin et al., 2024). For agent $i$, given a local policy $\pi_i = \{\pi_{i,h}\}_{h \in [H]}$ and a public context $c$, the transition of the local state is indeed independent from other agents and thus we can define the local state visitation measure as follows:

$$d_h^{\pi_i}(s|c) := \mathbb{P}^{\pi_i}(s_{i,h} = s|c), \qquad \forall h \in [H], s \in \mathcal{S}_i, i \in [N],$$

where $s_{i,h}$ is the local state of agent $i$ at step $h$ and $\mathbb{P}^\pi(\cdot|c)$ denotes the distribution of the trajectories under policy $\pi_i$ and public context $c$. We also define $d_h^{\pi_i}(s, a|c) := d_h^{\pi_i}(s|c)\pi_{i,h}(a|s)$. We want to identify *a natural structural property* for MARL, under which we can design decentralized *statistically and computationally efficient offline* algorithm, which means that we assume access to an offline dataset $\mathcal{D}$ without allowing interaction with the environment beyond this.

## 3 Interaction Rank Implies Robustness to Distribution Shift

In this section, we define the key structural property introduced in this work—the *interaction rank* (IR) of a function. We show that a function with a low interaction rank is significantly more robust to distribution shift compared to a general function in a standard offline supervised learning setting. This observation later enables us to derive sample efficient guarantees for the MARL setting. For an arbitrary function, we define its interaction rank as follows.

**Definition 2** (Interaction Rank). *A function $f : \mathcal{X} \times \mathcal{Y}_1 \times \cdots \times \mathcal{Y}_W \to [0, 1]$ has interaction rank $K$ ($K$-IR) if there exists a positive integer $K$ such that there exists a group of sub-functions $\cup_{0 \leq k \leq K} \{g_{j_1,\ldots,j_k}\}_{j_1 < \cdots < j_k}$ which satisfies*

$$f(x, y_1, \ldots, y_W) = \sum_{k=0}^{K-1} \sum_{1 \leq j_1 < \cdots < j_k \leq W} g_{j_1,\ldots,j_k}(x, y_{j_1}, \ldots, y_{j_k}), \forall x \in \mathcal{X}, y_1 \in \mathcal{Y}_1, \ldots, y_W \in \mathcal{Y}_W.$$

Intuitively, the function can be decomposed into a sum of $g$, called *sub-functions*, each depending only on a subset of the input variables. This structure is common in practice and finds application in fields including physics (Grana, 2016), economics, (Asghari et al., 2022) and statistics (Vonesh et al., 2001).

**Relation to Taylor series.** When restricting the inputs of a function to a local neighborhood, Definition 2 can understood as a Taylor expansion of function. To see this, fix an $x \in \mathcal{X}$, then any $K$-differentiable function $f$ in a local region of $\{y_w\}_{w=1}^W \in \prod_w^W \mathcal{Y}_w$ can be approximated as

$$f(x, y_1, \ldots, y_W) \simeq f(x, y_1', \ldots, y_W') + \sum_{k=1}^K \frac{1}{k!} \sum_{j_1, \ldots, j_k} \frac{\partial f(x, y_1', \ldots, y_W')}{\partial y_{j_1} \cdots \partial y_{j_k}} \prod_{k'=1}^k (y_{j_{k'}} - y_{j_{k'}}').$$

Hence, the interaction rank of a $K$-order Taylor expansion is upper bounded by $K + 1$. Further, if the Taylor series is close to $f$, we can find a good approximation of $f$ with low interaction rank.

**Bounded interaction rank implies distribution shift robustness.** The key property that makes functions with low interaction rank useful in the offline MARL is their robustness to distribution shift. Towards formalizing this statement, let us first consider an offline supervised learning setting. Suppose we wish to learn a target function $f^\star$ in an offline setting. The training distribution is $x \sim p, y_i \sim p_i(\cdot|x), \forall i$ and the target distribution is $x \sim p', y_i \sim p_i'(\cdot|x), \forall i$. The distribution shift is quantified by the density ratio:

$$\max_{x \in \mathcal{X}} \frac{p'(x)}{p(x)} \le C_{\mathsf{DS}}, \quad \max_{i \in [N], x \in \mathcal{X}, y_i \in \mathcal{Y}_i} \frac{p_i'(y_i|x)}{p_i(y_i|x)} \le C_{\mathsf{DS}}.$$

Let $\widehat{f}$ denote the learned function. Standard guarantees imply that the training error (i.e., under $p$ and $p_i$) can be upper bounded by $\epsilon$:

$$\mathbb{E}_{x \sim p, y_1 \sim p_1(\cdot|x), \ldots, y_W \sim p_W(\cdot|x)} \left[ \left( (f^\star - \widehat{f})(x, y_1, \cdots, y_W) \right)^2 \right] \le \epsilon. \tag{1}$$

When $f^\star$ and $\widehat{f}$ are general functions, the optimal worst-case learning error under the *target distribution* is $O\big((C_{\mathsf{DS}})^{W+1}\epsilon\big)$, which scales exponentially with the input size $W$. However, if $f^\star$ and $\widehat{f}$ have bounded interaction rank, this result can be significantly improved; the error under distribution shift only scales exponentially with the interaction rank.

**Theorem 1.** *If $f^\star$ and $\widehat{f}$ are $K$-IR, we have*

$$\mathbb{E}_{x \sim p', y_1 \sim p_1'(\cdot|x), \ldots, y_W \sim p_W'(\cdot|x)} \left[ \left( (f^\star - \widehat{f})(x, y_1, \ldots, y_W) \right)^2 \right] \lesssim (2W)^{2(K-1)} C_{\mathsf{DS}}^K \epsilon.$$

Here for any two functions $g$ and $g'$, $g \lesssim g'$ means that there exists a constant $c > 0$ such that $g < cg'$ always holds. Theorem 1 indicates that when $K \ll W$, function classes with bounded interaction rank are more robust to distribution shift and can significantly alleviate the curse of dimensionality due to multiple agents in offline learning. In MARL, $W + 1$ will be the number of agents, while $K$ is the interaction rank of the reward.

## 4 WARM UP: CONTEXTUAL GAMES

The robustness to distribution shift of low-IR functions suggests that such a property may be useful for offline MARL. Indeed, in the offline setting we need to properly estimate quantities that deviate from the data distribution. To provide intuition for the benefits of low-IR reward classes and corresponding algorithmic design, we start by considering the *contextual games* (CG) setting as a warm up.

**Offline CG.** The CG problem is a general-sum contextual MG where $\mathcal{S}_i = \emptyset$ for all $i$ and $H = 1$. To simplify notation, we omit the $h$ subscript in $r_h$ and $\pi_h$ for this setting. We assume the offline dataset $\mathcal{D} = \mathcal{D}_{\mathsf{R}}$ where each sample $(c, \boldsymbol{a} = \{a_i\}_{i=1}^N, \{r_i\}_{i=1}^N)$ is i.i.d. sampled from $c \sim \rho, a_i \sim \nu_i(\cdot|c), r_i \sim R_i^\star(c, \boldsymbol{a})$ for all $i \in [N]$. We call $\nu_i$ the offline behavior policy for each agent $i$ and use $\nu$ to denote the product behavior policy $\prod_{i \in [N]} \nu_i$. Let us assume for simplicity that we have learned reward functions $\{\widehat{r}_i \in [0, 1]\}_{i \in [N]}$ from the offline dataset with in-distribution training error $\epsilon$:

$$\mathbb{E}_{c \sim \rho, \boldsymbol{a} \sim \nu(\cdot|c)} \left[ (r_i^\star - \widehat{r}_i)^2 \right] \le \epsilon, \quad \forall i \in [N].$$

**Algorithm: Decentralized $\chi^2$-Regularized Policy Gradient.** Given $\widehat{r}$, we propose a *decentralized, $\chi^2$-regularized, no-regret* policy gradient based algorithm. As we show, this algorithm produces a set of policies which are near equilibrium. In each iteration $t$, each agent will update their policy via:

$$\pi_i^{t+1}(c) = \underset{p \in \Delta_{\mathcal{A}_i}}{\arg\min} -\langle \widehat{r}_i^t(c, \cdot), p \rangle + \lambda \underbrace{\chi^2(p, \nu_i(c))}_{\text{regularization}} + \frac{1}{\eta} \underbrace{D_{c,i}(p, \pi_i^t(c))}_{\text{no-regret learning}}. \tag{2}$$

Here $\widehat{r}_i^t(c, a_i) = \mathbb{E}_{a_j \sim \pi_j^t(c), \forall j \neq i}[\widehat{r}_i(c, \boldsymbol{a})]$ is the expected reward of agent $i$ given that the other agents' policies are $\prod_{j \neq i} \pi_j^t$. The regularizer $\chi^2(p, \nu_i(c)) := \mathbb{E}_{a_i \sim \nu_i(\cdot|c)}[(p(a_i)/\nu_i(a_i|c) - 1)^2]$ is the $\chi^2$-divergence between distribution $p$ and $\nu_i(c)$ and $D_{c,i}(p, \pi_i^t(c))$ is the Bregman divergence between distribution $p$ and $\pi_i^t(c)$:

$$D_{c,i}(p, \pi_i^t(c)) := \chi^2(p, \nu_i(c)) - \chi^2(\pi_i^t(c), \nu_i(c)) - \langle \nabla_{\pi_i^t}(c) \chi^2(\pi_i^t(c), \nu_i(c)), p - \pi_i^t(c) \rangle$$

$$= \mathbb{E}_{a_i \sim \nu_i(\cdot|c)} \left[ \left( \frac{p(a_i) - \pi_i^t(a_i|c)}{\nu_i(a_i|c)} \right)^2 \right].$$

We denote the total number of iterations by $T$. Eq. (2) has two divergence terms, which serve different roles. We add the $\chi^2$-divergence regularization term to encourage the policy trajectory to stay close to the behavior policy $\nu_i$ and thus lessen the distribution shift issue. On the other hand, to ensure the update enjoys *no regret*, we have a Bregman divergence term which is motivated from the policy mirror descent literature (Zhan et al., 2023a; Lan, 2023). Notably, Eq. (2) is a quadratic optimization problem whose input size is only $|\mathcal{A}_i|$. Thus, for small action and state space we can solve it efficiently without incurring exponential computation cost as the number of agents increase.

**Remark 2.** *The key ingredients of our algorithm are (1) regularization and (2) no-regret learning. We choose $\chi^2$-divergence and its corresponding Bregman divergence for a tractable theoretical analysis. In practice, other regularizers can also be utilized, such as KL divergence (Rafailov et al., 2024) or the $L_2$ behavior cloning term in* `TD3-BC` *(Fujimoto and Gu, 2021). Additionally, in practice, one-step online gradient (Zinkevich, 2003) can be used as the no-regret learning algorithm.*

**Theoretical analysis.** Now we analyze the statistical sample complexity of the above algorithm. If the reward function class has no specific structure, the sample complexity can still scale exponentially with $N$ due to distribution shift. To address this, we leverage a *low-IR reward function class*:

**Assumption 1** ($K$-IR Reward). *Suppose that the interaction rank of $r_i^\star$ and $\widehat{r}_i$ are upper bounded by $K$, with $\mathcal{X} = \mathcal{C} \times \mathcal{A}_i$ and $\mathcal{Y}_j = \mathcal{A}_j$ in Definition 2 for all $i \in [N]$.*

Assumption 1 naturally holds in a variety of games. For example, polymatrix games (Howson Jr, 1972; Kalogiannis and Panageas, 2024; MacQueen and Wright, 2024) characterize the reward function via pairwise interactions and, thus, for these settings Assumption 1 holds with $K = 2$. In network games (Galeotti et al., 2010; DeWeese and Qu, 2024; Park et al., 2024), the reward only depends on the neighbors and thus Assumption 1 holds with $K$ equal to the degree of the network. Note that for all of these examples, we have $K \ll N$.

Now we introduce a bound on the maximum gap of the output policy $\widehat{\pi}$ under $K$-IR reward classes. Let $r(\pi)$ be the expected reward under the distribution $c \sim \rho, \boldsymbol{a} \sim \pi(\cdot|c)$. Similar to existing offline RL analysis techniques (Xie et al., 2021), we split the bound into on-support and off-support components:

**Theorem 2** (Informal). *Suppose Assumption 1 holds. Let $\Pi_i(C) := \{\mu_i : \mathbb{E}_{c \sim \rho}[\chi^2(\mu_i(c), \nu_i(c))] \leq C\}$ denote the policy class which has bounded $\chi^2$-divergence from the behavior policy $\nu_i$. Fix any $\delta \in (0, 1]$ and select $T, \eta, \lambda$ in Eq. (2) properly. Then, with probability at least $1 - \delta$, we have*

$$\max_i \mathsf{Gap}_i(\widehat{\pi}) \lesssim \max_{i \in [N]} \min_{C \geq 1} \left\{ C \left( (2N^2)^{K-1} \epsilon \right)^{\frac{1}{3K-1}} + \mathsf{subopt}_i(C, \widehat{\pi}) \right\}, \tag{3}$$

*where $\mathsf{subopt}_i(C, \widehat{\pi}) := \max_{\mu_i \in \Pi_i} r_i^\star(\mu_i, \widehat{\pi}_{-i}) - \max_{\mu_i \in \Pi_i(C)} r_i^\star(\mu_i, \widehat{\pi}_{-i})$ is the off-support bias.*

**Optimal bias-variance tradeoff.** We call $\Pi_i(C)$ a *covered policy class* because policies within it have bounded $\chi^2$-divergence from the behavior policy $\nu$, which implies that we can estimate their performance relatively accurately from the offline dataset. The right hand side of Eq. (3) can be viewed as a bias-variance decomposition of the gap. The first term is the variance term which

measures the distribution-shift effect of comparing against policies from $\Pi_i(C)$. The second term is the bias term which quantifies the performance difference between the global optimal policy and the optimal policy in the covered policy class. As $C$ increases, the considered covered policy class will expand and thus the variance term will grow while the bias term will diminish. Notably, our algorithm does not require any information about $C$ and the gap in Theorem 2 is upper bounded by the optimal $C$, which means that we can identify the best bias-variance tradeoff automatically.

**Polynomial sample complexity with single-agent concentrability.** Let us consider the following single-agent all-policy concentrability coefficient $C_{\sf sin} := \max_{i \in [N], \mu_i, c \in \mathcal{C}, a_i \in \mathcal{A}_i} \frac{\mu_i(a_i|c)}{\nu_i(a_i|c)}$. Note that $C_{\sf sin}$ will not scale with $N$ exponentially. Then Theorem 2 implies that if $C_{\sf sin} < \infty$, the maximum gap under the interaction rank structure can be upper bounded by

$$\max_i \mathsf{Gap}_i(\widehat{\pi}) \lesssim C_{\sf sin} \left( (2N^2)^{K-1} \epsilon \right)^{\frac{1}{3K-1}} .$$

Therefore, given a fixed $K$, we can learn an approximate CCE with *polynomial sample complexity* with respect to the number of agents $N$ under single-agent all-policy concentrability. This demonstrates the power of low-IR reward classes for MARL. When combined with regularization and no-regret learning, the sample complexity is significantly improved, making computationally- and statistically-efficient algorithm design possible in MARL.

**Proof highlights.** We provide a proof sketch of Theorem 2 for $K = 2$, supplying intuition for how $K$-IR reward classes benefit theoretical sample complexity. For any agent $i \in [N]$ and policy $\mu_i \in \Pi_i(C)$ where $C > 1$, we can bound the in-support gap $\sum_{t=1}^{T} \left( r_i^{\star}(\mu_i, \pi_{-i}^t) - r_i^{\star}(\pi^t) \right)$ as follows:

$$\underbrace{\sum_{t=1}^{T} \mathbb{E}_{c \sim \rho, a_i \sim \mu_i(c), \boldsymbol{a}_{-i} \sim \pi_{-i}^t}[(r_i^{\star} - \widehat{r}_i)(c, \boldsymbol{a})]}_{(1)} + \underbrace{\sum_{t=1}^{T} \mathbb{E}_{c \sim \rho, \boldsymbol{a} \sim \pi^t}[(\widehat{r}_i - r_i^{\star})(c, \boldsymbol{a})]}_{(2)} + \underbrace{\sum_{t=1}^{T} \left( \widehat{r}_i(\mu_i, \pi_{-i}^t) - \widehat{r}_i(\pi^t) \right)}_{(3)} .$$

We need to bound terms (1), (2), and (3). Term (3) is the performance difference when changing the policy of agent $i$ to $\mu_i$. Note that this is equivalent to the *regret* of agent $i$ with loss function $-\widehat{r}_i^t$ and thus we can bound it with similar techniques in policy mirror descent literature (Zhan et al., 2023a).

Term (1) represents the reward learning error under the comparator policy $\mu_i$ and learned policy $\pi_{-i}^t$, which is different from $\nu$. To control it, we need to tackle the *distribution shift* between the two. We use $g_{\emptyset}^i, \{g_j^i\}_{j \neq i}$ and $\widehat{g}_{\emptyset}^i, \{\widehat{g}_j^i\}_{j \neq i}$ to denote the decomposition of $r_i^{\star}$ and $\widehat{r}_i$, and use $\Delta_j^i$ to denote $g_j^i - \widehat{g}_j^i$. Since we apply a $K$-IR reward class Assumption 1, we can decompose term (1) as follows

$$\mathbb{E}_{c \sim \rho, a_i \sim \mu_i(\cdot|c), \boldsymbol{a}_{-i} \sim \pi_{-i}^t(\cdot|c)}[(r_i^{\star} - \widehat{r}_i)(c, \boldsymbol{a})]$$
$$= \mathbb{E}_{c \sim \rho, a_i \sim \mu_i(\cdot|c)} \left[ \Delta^i(c, a_i) \right] + \sum_{j \neq i} \mathbb{E}_{c \sim \rho, a_i \sim \mu_i(\cdot|c), a_j \sim \pi_j^t(\cdot|c)} \left[ \Delta_j^i(c, a_i, a_j) \right] .$$

Meanwhile, from the property of $\chi^2$-divergence, we have

$$\mathbb{E}_{c \sim \rho, a_i \sim \mu_i(\cdot|c), a_j \sim \pi_j^t(\cdot|c)} \left[ \Delta_j^i(c, a_i, a_j) \right]$$
$$\leq \sqrt{\mathbb{E}_{c \sim \rho, a_i \sim \nu_i(\cdot|c), a_j \sim \nu_j(\cdot|c)} \left[ \left( \Delta_j^i(c, a_i, a_j) \right)^2 \right] \cdot \left( 1 + \chi^2 \left( \rho \circ (\mu_i \times \pi_j^t), \rho \circ (\nu_i \times \nu_j) \right) \right)},$$

where we use $\rho \circ p$ to denote the joint distribution $c \sim \rho, a \sim p(\cdot|c)$ for some conditional distribution $p$. For the $\chi^2$-divergence term, $\chi^2(\mu_i(c), \nu_i(c))$ is bounded because $\mu_i$ is from the covered policy class; we can also upper bound $\chi^2(\pi_j^t(c), \nu_j(c))$ *due to the $\chi^2$ regularizer term in Eq.* (2). Thus, we only need to bound $\mathbb{E}_{c \sim \rho, a_i \sim \nu_i(\cdot|c), a_j \sim \nu_j(\cdot|c)} \left[ \left( \Delta_j^i(c, a_i, a_j) \right)^2 \right]$.

This is non-trivial because we are only regressing with respect to $r^{\star}$, which is the summation of the sub-function $g$, and there exist infinite number of IR decompositions of $r^{\star}$. Fortunately, we are able to show that such an *aligned* decomposition exists:

**Lemma 1** (Sub-function Alignment for $K = 2$, informal)**.** *There exists a standardized IR decomposition of $r^{\star}$ and $\widehat{r}$, denoted by $g_{\emptyset}, g_1, \ldots, g_W$ and $\widehat{g}_{\emptyset}, \widehat{g}_1, \ldots, \widehat{g}_W$ such that we have*

$$\mathbb{E}_{c \sim \rho, a_i \sim \nu_i(\cdot|c), a_j \sim \nu_j(\cdot|c)} \left[ \left( \Delta_j^i(c, a_i, a_j) \right)^2 \right] \leq 2\epsilon, \quad \forall j \neq i.$$

With Lemma 1, we are able to bound term (1) efficiently. Term (2) can be handled similarly. Notably, Lemma 1 holds for general $K$ as shown in Lemma 4 and *the IR decomposition circumvents exponential scaling with $N$*. The above discussion illustrates that low-IR reward classes are quite effective when mitigating the learning error under distribution shift in MARL.

## 5 DECENTRALIZED REGULARIZED ACTOR-CRITIC IN MARKOV GAMES WITH DECOUPLED TRANSITIONS

We are now ready to investigate the benefits of low interaction rank in offline MGs. In particular, we will propose our main algorithmic framework to utilize low-IR function classes.

**Offline dataset.** We assume access to an offline dataset $\{\mathcal{D}_h\}_{h=1}^H$. $\mathcal{D}_h$ consists of $M$ i.i.d. samples $(c, \{s_i, a_i, s_i'\}_{i \in [N]}, \{r_i\}_{i \in [N]})$ where $c \sim \rho$, $s_i \sim \sigma_{i,h}(\cdot|c)$, $a_i \sim \nu_{i,h}(\cdot|c, s_i)$, $s_i' \sim P_{i,h}^\star(\cdot|c, s_i, a_i)$ and $r_i \sim R_{i,h}^\star(c, \{s_i, a_i\}_{i \in [N]})$. Note that $\sigma_{i,h}$ may not be the local state visitation measure $d_h^{\nu_i}(\cdot|c)$. We also use $\sigma_h$ to denote $\prod_{i \in [N]} \sigma_{i,h}$.

**General function approximation.** We consider the general function approximation setting. This makes the algorithm applicable in potentially large or even infinite state space and action space. Suppose that we have function classes $\mathcal{R} = \{\mathcal{R}_i\}_{i=1}^N$ to approximate the reward function $r_{i,h}^\star$ where $\mathcal{R}_i \subseteq \{r : \mathcal{C} \times \mathcal{A} \to [0, 1]\}$ for all $i \in [N]$. In addition, we use function classes $\{\mathcal{P}_i\}_{i \in [N]}$ where $\mathcal{P}_i \subseteq \{P : \mathcal{C} \times \mathcal{S}_i \times \mathcal{A}_i \to \Delta_{\mathcal{S}_i}\}$ to approximate the transition model. We assume here that $\mathcal{R}_i$ and $\mathcal{P}_i$ are finite, but the analysis can be extended to infinite function classes naturally by replacing the cardinality of $\mathcal{R}_i$ and $\mathcal{P}_i$ with its covering or bracketing number (Wainwright, 2019). To simplify notation, we use $|\mathcal{R}|$ and $|\mathcal{P}|$ to denote $\max_{i \in [N]} |\mathcal{R}_i|$ and $\max_{i \in [N]} |\mathcal{P}_i|$.

### 5.1 ALGORITHMIC FRAMEWORK

---

**Algorithm 1** Decentralized Regularized Actor-Critic (`DR-AC`)

---

1: Initialize $\pi_i^1$ to be the behavior policy $\nu_i$ for each agent $i$.
2: **/** Offline Reward & Transition Learning **/**
3: Compute for all $i \in [N], h \in [H]$

$$\widehat{r}_{i,h} = \arg\min_{r \in \mathcal{R}_i} \sum_{(c, \boldsymbol{s}, \boldsymbol{a}, r_i) \in \mathcal{D}_h} (r(c, \boldsymbol{s}, \boldsymbol{a}) - r_i)^2, \ \widehat{P}_{i,h} = \arg\max_{P \in \mathcal{P}_i} \sum_{(c, s_i, a_i, s_i') \in \mathcal{D}_h} \log P(s_i'|c, s_i, a_i).$$

4: **for** $t = 1, \ldots, T$ **do**
5:     **for** $i \in [N], h \in [H]$ **do**
6:         **/** Critic Update **/**
7:         Estimate the single-agent Q-function with the learned reward $\widehat{r}_i$ and transition $\widehat{P}_i$:

$$\widehat{Q}_{i,h}^t(c, s_i, a_i) = \mathbb{E}_{(s_j, a_i) \sim \widehat{d}_h^{\pi_j}(\cdot|c), \forall j \neq i} \left[ \widehat{Q}_{i,h}^{\pi^t, \widehat{r}}(c, \boldsymbol{s}, \boldsymbol{a}) \right], \forall c \in \mathcal{C}, s_i \in \mathcal{S}_i, a_i \in \mathcal{A}_i.$$

8:         **/** Actor Update **/**
9:         Run mirror descent for all $c \in \mathcal{C}, s \in \mathcal{S}_i$:

$$\pi_{i,h}^{t+1}(c, s) = \arg\min_{p \in \Delta_{\mathcal{A}_{i,h}}} -\langle \widehat{Q}_{i,h}^t(c, s, \cdot), p \rangle + \lambda \chi^2(p, \nu_{i,h}(c, s)) + \frac{1}{\eta} D_{c,s,i}(p, \pi_{i,h}^t(c, s)). \quad (4)$$

10: **Return:** the uniform mixture of $\left\{ \prod_{i \in [N]} \pi_i^t \right\}_{t=1}^T$.

---

For general-sum MGs with decoupled transitions, we consider a widely-used kind of algorithmic framework in practice, the actor-critic method (Barto et al., 1983). Arming it with regularization and no-regret learning, we propose `DR-AC` for offline learning in MGs. The full algorithm is stated in Algorithm 1. Notably, `DR-AC` is a *decentralized* model-based algorithm which is *computationally efficient given that we are able to solve a least squares regression (LSR) and maximum likelihood estimation (MLE) problem*. `DR-AC` consists of two phases: offline reward and transition learning, followed by decentralized actor-critic updates.

**Offline reward and transition learning.** We first learn the reward function $\widehat{r}_i$ for each agent $i$ using LSR on the offline dataset $\mathcal{D}_{\mathsf{R}}$. In particular, here we will use a function class $\mathcal{R}_i$ where all the functions have bounded IR so that our learned reward has higher robustness to distribution shift, as we have shown in the previous section. We also learn the transition model for each $i$ via MLE on the offline dataset with function classes $\mathcal{P}_i$. Note that LSR and MLE problems are common in supervised learning and can be solved with simple methods like stochastic gradient descent (Jain et al., 2018). The RL literature has also assumed the existence of efficient solutions to these optimization problems, calling algorithms that depend on them *oracle-efficient* (Dann et al., 2018; Agarwal et al., 2020; Uehara et al., 2021; Song et al., 2022).

**Critic update.** In each iteration, for each agent $i$, we estimate its current single-agent Q-function, given other agents' policies, with the learned reward $\widehat{r}$ and transition model $\widehat{P}$:

$$\widehat{Q}_{i,h}^t(c, s_i, a_i) = \mathbb{E}_{(s_j, a_j) \sim \widehat{d}_h^{\pi_j}(\cdot|c), \forall j \neq i} \left[ \widehat{Q}_{i,h}^{\pi^t, \widehat{r}}(c, \boldsymbol{s}, \boldsymbol{a}) \right],$$

where we use $\widehat{Q}_{i,h}^{\pi, \widehat{r}}$ and $\widehat{d}_h^{\pi_j}$ to denote the joint Q-function and local state visitation measure of $\pi$ under reward $\widehat{r}$ and transition $\widehat{P}$. In practice, we can simply use a Monte-Carlo-type method to estimate $\widehat{Q}_{i,h}^t$, which only requires solving an LSR problem and is thus computationally efficient. See Appendix C for more details.

**Actor update.** Given the estimated Q-function, we use regularized policy gradient to update each agent's policy. The update formula Eq. (4) is almost the same as the update in Eq. (2) for CGs, except the estimated reward is replaced with the estimated Q-function. We use $\chi^2$-divergence for regularization and Bregman divergence in Algorithm 1. Nevertheless, DR-AC allows other regularizers and no-regret learning techniques as mentioned in Remark 2. Note that Eq. (4) is a quadratic optimization problem with input size $|\mathcal{A}_i|$ and thus can be solved efficiently.

## 5.2 THEORETICAL ANALYSIS

We now present the sample complexity guarantee for DR-AC. We assume the function class $\{\mathcal{R}_i\}_{i \in [N]}$ and $\{\mathcal{P}_i\}_{i \in [N]}$ are realizable.

**Assumption 2.** *Suppose that we have $r_{i,h}^\star \in \mathcal{R}_i$ and $P_{i,h}^\star \in \mathcal{P}_i$ for all $i \in [N], h \in [H]$.*

In general, DR-AC can have exponentially large statistical complexity with respect to the number of agents $N$. However, similarly to the CG result, a low-IR reward function class alleviates this.

**Assumption 3** (K-IR Reward). *Suppose that the IR of $r_i$ is upper bounded by $K$ with $\mathcal{X} = \mathcal{C} \times \mathcal{S}_i \times \mathcal{A}_i$ and $\mathcal{Y}_j = \mathcal{S}_j \times \mathcal{A}_j$ in Definition 2 for all $j \neq i, r_i \in \mathcal{R}_i, i, j \in [N]$.*

In addition, we assume that the offline dataset satisfies *single-agent* all-policy concentrability for the local state distribution. Recall that $\sigma_{i,h}$ is the dataset distribution.

**Assumption 4.** *Suppose that for all $i \in [N]$ we have $\max_{\mu_i, c \in \mathcal{C}, s \in \mathcal{S}_i, h \in [H]} \frac{d_h^{\mu_i}(s|c)}{\sigma_{i,h}(s|c)} \leq C_{\mathsf{S}} < \infty$.*

We need Assumption 4 because bounded $\chi^2$-divergence between the action probabilities of two policies does not imply bounded $\chi^2$-divergence between their state visitation measure. In DR-AC we can only regularize the action probability and therefore require additional concentrability for the local states. Nevertheless, here we only need single-agent concentrability and thus $C_{\mathsf{S}}$ does not scale exponentially with $N$. Now we can bound on the maximum gap of the output policy $\widehat{\pi}$ by DR-AC:

**Theorem 3.** *Suppose Assumption 2, Assumption 3 and Assumption 4 hold. Let $\Pi_i(C) := \{\mu_i : \mathbb{E}_{c \sim \rho, s \sim d_h^{\mu_i}(\cdot|c)}[\chi^2(\mu_{i,h}(c, s), \nu_{i,h}(c, s))] \leq C, \forall h\}$ denote the policy class which has bounded $\chi^2$-divergence from the behavior policy $\nu_i$. Fix any $\delta \in (0, 1]$ and select*

$$\lambda = C_{\mathsf{S}}^{\frac{K}{3K+2}} H^{\frac{3K}{3K+2}} (2N^2)^{\frac{K-1}{3K+2}} \epsilon_{\mathsf{RP}}^{\frac{1}{3K+2}}, \quad \eta = \frac{\lambda}{H^2}, \quad T = \frac{H^2}{\lambda^2},$$

*where $\epsilon_{\mathsf{RP}} := \frac{\log(NH|\mathcal{R}||\mathcal{P}|/\delta)}{M}$. Then, with probability at least $1 - \delta$ the output of DR-AC, $\widehat{\pi}$, satisfies:*

$$\max_i \mathsf{Gap}_i(\widehat{\pi}) \lesssim \max_{i \in [N]} \min_{C \geq 1} \left\{ C C_{\mathsf{S}}^{\frac{K}{3K+2}} H^{\frac{6K+2}{3K+2}} (2N^2)^{\frac{K-1}{3K+2}} \epsilon_{\mathsf{RP}}^{\frac{1}{3K+2}} + \mathsf{subopt}_i(C, \widehat{\pi}) \right\},$$

*where $\mathsf{subopt}_i(C, \widehat{\pi}) := \max_{\mu_i} \mathbb{E}_{c \sim \rho}[V_{i,1}^{\mu_i \circ \widehat{\pi}_{-i}}(c, \boldsymbol{s}_1)] - \max_{\mu_i \in \Pi_i(C)} \mathbb{E}_{c \sim \rho}[V_{i,1}^{\mu_i \circ \widehat{\pi}_{-i}}(c, \boldsymbol{s}_1)].$*

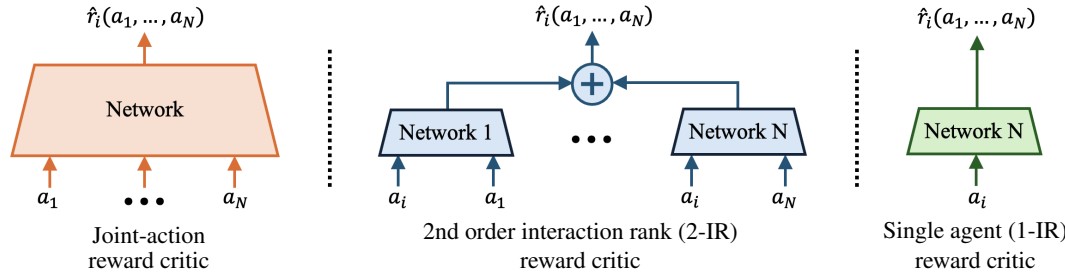

Figure 1: Network diagrams for the $i$th agent.

Similarly to Theorem 2, Theorem 3 indicates that `DR-AC` admits an optimal bias-variance tradeoff over the covered policy class $\Pi_i(C)$. In addition, if we have single-agent all-policy concentrability $C_{\sf sin} := \max_{h,i,\mu_i,c,s_i \in \mathcal{S}_i, a_i \in \mathcal{A}_i} \frac{\mu_{i,h}(a_i|c,s_i)}{\nu_{i,h}(a_i|c,s_i)} < \infty$, `DR-AC` is capable of learning an $\epsilon$-approximate CCE given sample complexity

$$M \gtrsim \frac{C_{\sf sin}^{3K+2} C_{\sf S}^K H^{6K+2} (2N^2)^{K-1} \log(NH|\mathcal{R}||\mathcal{P}|/\delta)}{\epsilon^{3K+2}},$$

Therefore, given a fixed $K$ and single-agent all-policy concentrability, `DR-AC` can learn an approximate CCE in polynomial sample complexity with respect to $N$ for general-sum MGs with decoupled transitions. This suggests that introducing low-IR structure to the reward class is still beneficial for offline learning in general-sum MGs.

**Remark 3.** *For $|\mathcal{R}|$, note that we require the function classes $\mathcal{R}_i$ to have IR bounded by $K$. This means that their complexity will at most only scale with $K$ exponentially.*

**Comparison with existing works.** To our knowledge, Cui and Du (2022a); Zhang et al. (2023b) are the only existing offline general-sum MARL works with provable statistical guarantees. However, the proposed methods are not decentralized and require evaluating the gap for *every possible candidate joint policies*, resulting in an impractically high computational burden.

Statistically, although Cui and Du (2022a) achieves $\widetilde{O}(1/\epsilon^2)$ complexity, they require stronger concentrability assumption, which is the following unilateral concentrability with a target policy $\pi^\star$:

$$C_{\sf uni}(\pi^\star) := \max_{h,i,\mu_i,c,\boldsymbol{s} \in \mathcal{S}, \boldsymbol{a} \in \mathcal{A}} \frac{d_h^{\mu_i}(s_i, a_i|c) d_h^{\pi^\star_{-i}}(\boldsymbol{s}_{-i}, \boldsymbol{a}_{-i}|c)}{\sigma_h(\boldsymbol{s}|c)\nu(\boldsymbol{a}|\boldsymbol{s}, c)},$$

where $\boldsymbol{s}_{-i}$ and $\boldsymbol{a}_{-i}$ are the joint state and action of the agents excluding $i$. Note that the single-agent all-policy concentrability coefficient $C_{\sf S}C_{\sf sin}$ is indeed weaker than $C_{\sf uni}$ and we have $C_{\sf S}C_{\sf sin} \le C_{\sf uni}(\pi^\star)$ for any $\pi^\star$. In the worst case, $C_{\sf uni}$ can still scale exponentially with number of agents $N$, whereas our sample complexity scales with the IR $K$.

For Zhang et al. (2023b), in CGs, they have the following concentrability assumption:

$$C'_{\sf uni}(\mathcal{R}, \pi^\star) := \max_{i,\mu_i,r \in \mathcal{R}_i} \frac{\mathbb{E}_{c \sim \rho, a_i \sim \mu_i(\cdot|c), \boldsymbol{a}_{-i} \sim \pi^\star_{-i}(\cdot|c)}[(r - r^\star)^2]}{\mathbb{E}_{c \sim \rho, \boldsymbol{a} \sim \nu(\cdot|c)}[(r - r^\star)^2]}.$$

In their work $\mathcal{R}_i$ can be a general function class and thus $C'_{\sf uni}(\mathcal{R}, \pi^\star)$ can be as large as $C_{\sf uni}(\pi^\star)$ in the worst case. Notably, if we use function class with $K$-IR instead, Theorem 1 shows that $C'_{\sf uni}(\mathcal{R}, \pi^\star) \lesssim C_{\sf sin}^K$. Therefore, we indeed find a particular function class such that the concentrability in Zhang et al. (2023b) is not vacuous. For MGs, Zhang et al. (2023b) uses a function class to approximate the *joint* Q function while we use $\mathcal{F}$ to approximate the *single-agent* Q-function, and thus the results are not directly comparable.

## 6 EXPERIMENTS

In this section, we examine the practical implications of our results. With this in mind, our findings can be interpreted as providing the following guideline: *Use a reward or Q-function class with the smallest possible IR that can still represent the underlying true model.* This approach strikes a balance between two factors: it ensures realizability by requiring the model can be represented accurately, and it improves sample efficiency, as demonstrated in Theorem 2 and Theorem 3.

**Implementation and experimental setting.** To examine the usefulness of this observation, we study a simple offline CG environment. We implement the actor update in `DR-AC` to be a single gradient descent update with respect to `TD3+BC` objective (Fujimoto and Gu, 2021) from Tianshou library (Weng et al., 2022). Further, recall that `TD3+BC` adds explicit $L_2$ regularization term that keeps the policy close to the data collection policy and thus fits into the framework of `DR-AC`. To test the potential benefits of low rank reward critic architectures we experimented with three different types, depicted in Figure 1: i) joint-action, ii) 2-IR, and iii) 1-IR reward critics. The joint-action reward critic is a general mapping from the joint action space to a number, and, hence, is the most expressive; it can represent both 2-IR and 1-IR. On the other hand, the 1-IR architecture is the least expressive, as it cannot represent 2-IR reward models, since it only accesses a single agent action. Notably, we choose the number of parameters of the 2-IR and joint-action architectures to be of the same order of magnitude for fair comparison.

The details of our environment setting are as follows (see Appendix B for additional information). We consider the continuous action setting, where $\forall i \in [N]$, $a_i \in [-1, 1]$. The underlying reward model is a 2-IR function of the form $\forall i \in [N]$, $r_i^\star(\boldsymbol{s}, \boldsymbol{a}) = \sum_{j=1}^{N} a_i a_j / \sqrt{N} + \epsilon$ where

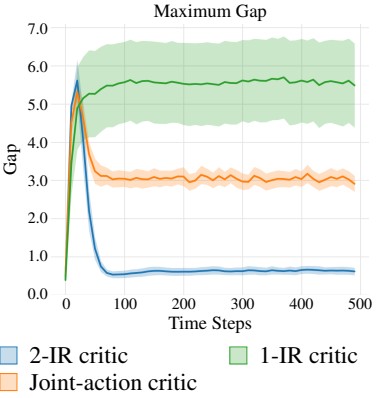

Figure 2: Comparison of `TD3+BC` instantiated with different critic architectures, i) 1-IR critic, ii) 2-IR critic, and iii) joint-action critic. The underlying true reward is a 2-IR. This figure showcases the advantage of using 2-IR critic architecture compared to 1-IR or the general joint-action critics when the underlying model is 2-IR. The shaded area represents the standard error across trials.

$\epsilon \sim \text{Uniform}(-\sigma, \sigma)$ and $\sigma > 0$. Further, we set number of agents as $N = 50$. We collect offline data with the uniform policy and set the number of samples $M$ such that $\sigma N/M = 0.1$. In this noise regime, the reward model is learnable but the noise level may effect the training procedure. We experiment with few architectures for each reward critic type and report here the best one. We also experimented with an additional environment in which the underlying reward is a 1-IR model (see additional results in Appendix B).

**Results.** Experiment results are depicted in Figure 2. The 2-IR critic approach leads to the best performing result by significant margin compared to the joint-action and 1-IR reward critics. For the 2-IR critic the maximum gap across agents is the smallest, meaning the joint policy is in a near equilibrium point. Interestingly, the simpler 1-IR model has the worst performance among the three candidates. Such an approach for critic modeling is common in the online cooperative MARL setting (Sunehag et al., 2017; Rashid et al., 2020; Yu et al., 2022). Nevertheless, as our experiments show, it can dramatically fail in offline MARL. This is because in the online setting, the agent can continually collect fresh samples to update the estimated 1-IR reward so that the critic can learn accurate *local* approximations of the current expected reward even if the other agents' policies change. However, in offline setting, a 1-IR critic cannot make such updates because iterative data collection is not allowed. In the offline MARL setting, single agent critic models may be severely biased and degrade the performance of the learned policies.

## 7  CONCLUSIONS

In this work, we investigated the benefits of using reward models with low IR in the offline MARL setting. We showed that learning an approximate equilibrium in offline MARL can scale exponentially with the IR instead of exponentially with the number of agents. Our proposed algorithm is a decentralized, no-regret learning algorithm that can be implemented in practical settings while utilizing standard RL algorithms. The empirical results demonstrate superior performance of the critic with the smallest IR that can still represent the underlying true model in offline MARL, while the widely-used single-agent critic can fail catastrophically in this setting. Moving forward, building critics with low IR in MARL is a promising direction for future work, as well as exploring additional structural assumptions to alleviate the MARL problem.

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

# A  ADDITIONAL RELATED WORKS

We would like to point out that for offline general-sum Markov games, there are no existing decentralized and provably efficient algorithms and this field is under-explored. Our work can be viewed as a first attempt towards this objective. For more background information, the related works are stated in the following.

**Provable offline MGs.**  Cui and Du (2022b) studies offline tabular zero-sum Markov games and Yan et al. (2022) improves the complexity to minimax optimal. Zhong et al. (2022); Xiong et al. (2022) further extend the discussion to linear function approximation. However, the above works are all limited to two-agent zero-sum games. For multi-agent general-sum Markov games, the only existing works are Cui and Du (2022a); Zhang et al. (2023b), where Cui and Du (2022a) investigates the tabular setting and later Zhang et al. (2023b) proposes a centralized algorithm with function approximation. Nevertheless, both of them are centralized and computationally inefficient since they need to evaluate each possible joint policy, whose complexity will scale with the number of agents exponentially. In comparison, our method is decentralized and computationally feasible.

**Decentralized algorithms in general-sum MGs.**  In the online setting, to learn a CCE for a general-sum MG, Jin et al. (2021); Song et al. (2021); Mao and Başar (2023) proposed V-learning and Daskalakis et al. (2023) introduced SPoMAR. However, these algorithms cannot achieve no-regret and depend heavily on collecting fresh samples, thus cannot be extended to the offline setting trivially. For specific no-regret decentralized algorithms, as far as we know, Erez et al. (2023) is the only existing decentralized work which achieves sublinear regret in general-sum MGs (and thus is able to learn a CCE). Nevertheless, they only focus on tabular cases in the full information setting or online setting with a minimum reachability assumption, which are drastically different from the offline setting.

**Factored critics.**  Another closely related line of work is the utilization of factored critics in empirical studies. Since Sunehag et al. (2017) proposed the value decomposition networks (VDN), factored critics have been widely applied in the online cooperative MARL setting. Rashid et al. (2018) further proposed QMIX, a Q-learning type algorithm which utilizes a non-linear mixing network to combine single-agent critics, and Peng et al. (2021) extended the discussion into actor-critic methods. Nevertheless, most of these works focus on online cooperative MARL and lack theoretical analysis. Our paper indeed formalizes and generalizes these efforts by introducing the interaction-rank structure. As a matter of fact, VDNs are exactly a 1-IR critic. We further show that the idea of factorization in critics is also useful in offline general-sum MGs, but in general you need critics with a higher IR than VDNs for offline learning, as discussed in Section 6.

# B  ADDITIONAL EXPERIMENTAL DETAILS

In this section we give additional information on the experiment design. Additional hyper-parameters related to training are given in Table 2.

Our high-level implementation follows the framework of `DR-AC` and has three steps:

1. **Data collection.** Collect data via a uniform policy, where each agent executes a random action $a_i \sim \text{Uniform}([-1, 1])$ for all $i \in [N]$.

2. **Learn critic.** Learn $N$ reward critic models using LSR and the collected offline data. Namely, for each agent $i \in [N]$, estimate a reward critic by solving the following LSR:

$$\arg\min_{r \in \mathcal{R}_i} \sum_{(\boldsymbol{a}, r_i) \in \mathcal{D}_h} (r(\boldsymbol{a}) - r_i)^2.$$

   We experiment with three types of reward critic types, namely, different reward classes $\mathcal{R}_i$: 1-IR, 2-IR, and joint-action critic models. We solve this by gradient descent, which iteratively samples a batch from $\mathcal{D}_h$, and takes a gradient step. Our method returns the critic with the smallest validation loss, calculated with respect to a holdout validation dataset, through the course of training. Lastly, if during the run the critic does not show improvement after number of steps specified by the 'patience' parameter we stop the run (see Table 2 for hyper-parameter values).

3. **Learn actor.** Apply `TD3+BC` on all agents to get a policy per agent.

Next we elaborate on the critic architectures we used and their implementation.

1. **Joint-action critic.** We experimented with architectures with 3 layers and 2 layers. Recall that $N$ is the number of agents. The 3 layer architectures are of size $N \times \text{width} \times \text{width} \times \text{width} \times 1$ where $\text{width} \in [512, 1028, 2056]$, and the 2 layer architectures are of size $N \times \text{width} \times \text{width} \times 1$ where $\text{width} \in [128, 512, 2056]$.

2. **2-IR critic.** We experimented with 2 layer architectures of size $2 \times \text{width} \times \text{width} \times 1$ where $\text{width} \in [64, 128, 256]$. For the $i^{\text{th}}$ agent, there are $N$ such networks, where each network represents the interaction term with the $j^{\text{th}}$ agent. Let this network be denoted as $\text{DNN}_{ij} : \mathcal{A} \times \mathcal{A} \to \mathbb{R}$. With these, the reward of the $i^{\text{th}}$ agent is given by

$$\hat{r}_i(\boldsymbol{a}) = \sum_j \text{DNN}_{ij}(a_i, a_j).$$

3. **1-IR critic.** We experimented with 2 layer architectures of size $1 \times \text{width} \times \text{width} \times 1$ where $\text{width} \in [128, 256, 512]$, where the only input to the network is the action of the $i^{\text{th}}$ agent.

The metric which we measure is the maximum gap defined by

$$\max_{i \in [N]} \max_{a_i \in [-1, 1]} a_i \left( a_i + \sum_{j \neq i} \pi_j \right) - \pi_i \left( \sum_{j \in [N]} \pi_j \right),$$

| Hyperparameter | Value |
|---|---|
| Critic learning rate | 1e-4 |
| Critic batch size | 64 |
| Patience parameter for critic | 20 |
| Actor learning rate | 1e-3 |
| Actor batch size | 64 |
| Number of epochs | 500 |
| Optimizer | Adam |
| Policy architecture | MLP, 3 layers, width 128, w/ ReLu activations |
| `TD3+BC` $\alpha$ parameter | 5 |
| # of trials per experiment | 10 |

Table 2: Hyperparameters used in the experiments.

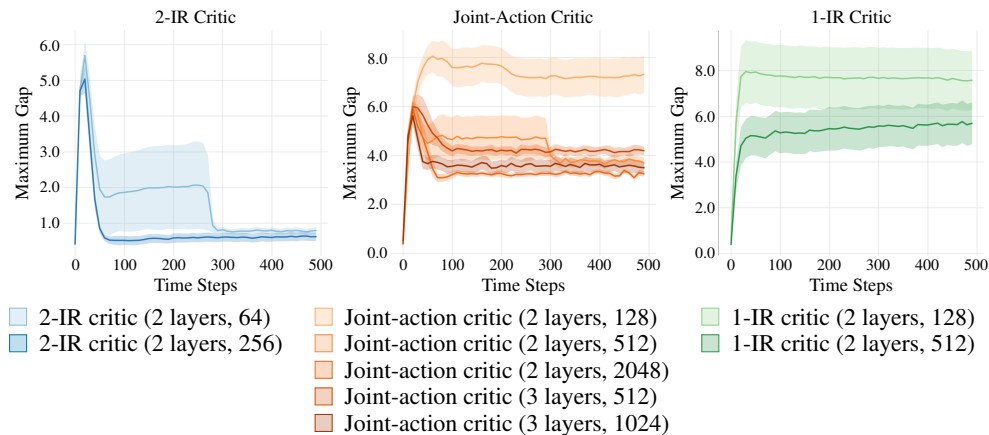

Figure 3: Comparison of `TD3+BC` instantiated with different critic architectures, i) 1-IR critic, ii) 2-IR critic, and iii) joint-action critic. The underlying true reward is a 2-IR. The shaded area represents the standard error computed across trials.

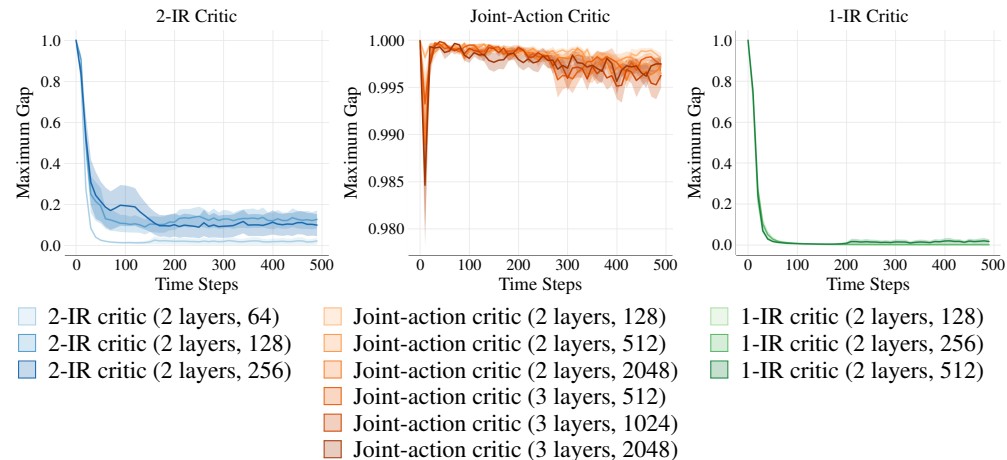

Figure 4: Comparison of `TD3+BC` instantiated with different critic architectures, i) 1-IR critic, ii) 2-IR critic, and iii) joint-action critic. The underlying true reward is a 1-IR. The shaded area represents the standard error computed across trials.

where $\pi_j$ is the policy for agent $j$ (note that here we use deterministic policies). In particular, the above expression obtains its maximum at $a_i = \pm 1$.

Details of the environment with the underlying reward of 2-IR are presented in Section 6. Figure 3 depicts additional results that measure the performance of various architectures for the 2-IR environment. As observed, the 2-IR critic consistently performs better compared to the joint-action architecture and the 1-IR architecture.

We experimented with an additional environment in which the underlying reward model is a 1-IR reward model of the form $\forall i \in [N]$, $r_i^\star(\boldsymbol{s}, \boldsymbol{a}) = a_i^2 + \epsilon$. Additional parameters of the environment are similar to those described in Section 6. Since the underlying reward model is a 1-IR, we expect the 1-IR critic type to result in good performance. Further, since the 2-IR critic is not significantly more expressive compared to the 1-IR critic, we may expect it to have good performance as well. Figure 4 depicts the results of this experiment for all reward critic types and architectures. These show that both the 1-IR and 2-IR reward critics have good performance, whereas the joint-action critic performs significantly worse with respect to the maximum gap metric.

## C  Q-FUNCTION ESTIMATION

In this section we provide a computationally efficient method to estimate $\widehat{Q}_{i,h}^t$ in Algorithm 1. We assume access to a function class $\{\mathcal{F}_i\}_{i \in [N]}$ where $\mathcal{F}_i \subseteq \{f : \mathcal{C} \times \mathcal{S}_i \times \mathcal{A}_i \to [0, H]\}$ to approximate the single-agent Q-functions. The full algorithm is shown in Algorithm 2.

Specifically, in Algorithm 2, we will sample $c \sim \rho, s_{i,h} \sim \sigma_{i,h}(\cdot|c), a_{i,h} \sim \frac{1}{2}\nu_{i,h}(\cdot|c, s_{i,h}) + \frac{1}{2}\pi_{i,h}(\cdot|c, s_{i,h}), (\boldsymbol{s}_{-i,h}, \boldsymbol{a}_{-i,h}) \sim \widehat{d}_h^{\pi^t_{-i}}(\cdot|c)$ and then roll out the joint policy $\pi^t$ in $\widehat{P}$. It can be observed that the cumulative reward $q$ is indeed an unbiased estimate of $\widehat{Q}_{i,h}^t(c, s_{i,h}, a_{i,h})$. Notably, we sample the state $s_{i,h}$ from the offline dataset to leverage the offline information. We also sample $a_{i,h}$ from $\frac{1}{2}\nu_{i,h} + \frac{1}{2}\pi_{i,h}^t$ such that the actions can cover the current policy $\pi_{i,h}^t$ and the competing policy $\mu_{i,h}$, which has bounded $\chi^2$-divergence from $\nu_{i,h}$. Then we only need to run LSR on the collected batch to estimate the Q-function. In summary, we can see that Algorithm 2 can be implemented with LSR oracles.

---

**Algorithm 2** Q-function Estimation

---

1: **Input**: Estimated reward $\widehat{r}$, estimated transition $\widehat{P}$, policy $\pi^t$, step $h$, agent $i$, function class $\mathcal{F}_i$.
2: $\mathcal{D}_{\sf sim} \leftarrow \emptyset$.
3: **for** $m = 1, \ldots, M_{\sf sim}$ **do**
4:     Sample $c \sim \rho$.
5:     Execute $\pi^t$ in $\widehat{P}$ with public context $c$ until step $h$.
6:     Denote the current joint local state excluding agent $i$ by $\boldsymbol{s}_{-i,h}$. Reset the state of agent $i$ to be $s_{i,h} \sim \sigma_{i,h}(\cdot|c)$.
7:     Execute $a_{i,h} \sim \frac{1}{2}\nu_{i,h}(\cdot|c, s_{i,h}) + \frac{1}{2}\pi_{i,h}^t(\cdot|c, s_{i,h})$ and $\boldsymbol{a}_{-i,h} \sim \pi_{-i}^t(\cdot|c, \boldsymbol{s}_{-i,h})$.
8:     Continue to execute the joint policy $\pi^t$ in $\widehat{P}$ until step $H$.
9:     Compute the cumulative reward staring from $(c, \boldsymbol{s}_h, \boldsymbol{a}_h)$ by $q$ under the reward model $\widehat{r}$. Add $(c, s_{i,h}, a_{i,h}, q)$ into $\mathcal{D}_{\sf sim}$.
10: Run LSR: $\widetilde{Q}_{i,h}^t = \arg\min_{f \in \mathcal{F}_i} \sum_{(c, s_{i,h}, a_{i,h}, q) \in \mathcal{D}_{\sf sim}} (f(c, s_{i,h}, a_{i,h}) - q)^2$.
11: **Return:** $\widetilde{Q}_{i,h}^t$.

---

### C.1  THEORETICAL GUARANTEE

Next we want to show that the estimated $\widetilde{Q}_{i,h}^t$ is close to $\widehat{Q}_{i,h}^t$. We have the following lemma:

**Lemma 2** (Q-function estimation error). *Suppose $\widehat{Q}_{i,h}^t \in \mathcal{F}_i$ for all $t, i, h$ and Assumption 4 holds. With probability at least $1 - \delta$, we have for all $i, t, h, \mu_i \in \Pi_i(C)$ that*

$$\mathbb{E}_{c \sim \rho, s_i \sim d_h^{\mu_i}(\cdot|c)} \left[ \left\langle \widehat{Q}_{i,h}^t(c, s_i, \cdot) - \widetilde{Q}_{i,h}^t(c, s_i, \cdot), \mu_{i,h}(\cdot|c, s_i) - \pi_{i,h}^t(\cdot|c, s_i) \right\rangle \right]$$
$$\lesssim \sqrt{\frac{CC_{\sf S}H^2 \log(NTH|\mathcal{F}|/\delta)}{M_{\sf sim}}}.$$

Recall that for any two functions $g$ and $g'$, $g \lesssim g'$ means that there exsits a constant $c > 0$ such that $g < cg'$ always holds. Note that from the proof of Theorem 3, Lemma 2 suggests that we can use $\widetilde{Q}_{i,h}^t$ as a surrogate of $\widehat{Q}_{i,h}^t$ and Theorem 3 still holds as long as $M_{\sf sim} \gtrsim \frac{CC_{\sf S}H^2 \log(NTH|\mathcal{F}|/\delta)}{\epsilon^2}$. Therefore, Algorithm 2 is indeed a computationally and statistically efficient Q-function estimator.

**Proof of Lemma 2.** From the guarantee of LSR (Lemma 13), we know with probability at least $1 - \delta$ that for all $i \in [N], t \in [T], h \in [H]$

$$\mathbb{E}_{c \sim \rho, s_i \sim \sigma_{i,h}(\cdot|c), a_i \sim \frac{1}{2}\nu_{i,h}(\cdot|c, s_{i,h}) + \frac{1}{2}\pi_{i,h}^t(\cdot|c, s_{i,h})} \left[ \left( \widehat{Q}_{i,h}^t(c, s_i, a_i) - \widetilde{Q}_{i,h}^t(c, s_i, a_i) \right)^2 \right]$$
$$\lesssim \frac{H^2 \log(NTH|\mathcal{F}|/\delta)}{M_{\sf sim}}.$$

Therefore, from Cauchy-Schwartz, we have

$$\mathbb{E}_{c\sim\rho,s_i\sim d_h^{\mu_i}(\cdot|c),a_i\sim\pi_{i,h}^t(\cdot|c,s_i)}\left[\left|\widehat{Q}_{i,h}^t(c,s_i,a_i)-\widetilde{Q}_{i,h}^t(c,s_i,a_i)\right|\right]\lesssim\sqrt{\frac{C_{\mathsf{S}}H^2\log(NTH|\mathcal{F}|/\delta)}{M_{\mathsf{sim}}}}.$$

On the other hand, since $\mu_i\in\Pi_i(C)$, from Lemma 5 we know

$$\mathbb{E}_{c\sim\rho,s_i\sim d_h^{\mu_i}(\cdot|c),a_i\sim\mu_{i,h}(\cdot|c,s_i)}\left[\left|\widehat{Q}_{i,h}^t(c,s_i,a_i)-\widetilde{Q}_{i,h}^t(c,s_i,a_i)\right|\right]\lesssim\sqrt{\frac{CC_{\mathsf{S}}H^2\log(NTH|\mathcal{F}|/\delta)}{M_{\mathsf{sim}}}}.$$

Therefore we have

$$\mathbb{E}_{c\sim\rho,s_i\sim d_h^{\mu_i}(\cdot|c)}\left[\left\langle\widehat{Q}_{i,h}^t(c,s_i,\cdot)-\widetilde{Q}_{i,h}^t(c,s_i,\cdot),\mu_{i,h}(\cdot|c,s_i)-\pi_{i,h}^t(\cdot|c,s_i)\right\rangle\right]$$
$$\lesssim\sqrt{\frac{CC_{\mathsf{S}}H^2\log(NTH|\mathcal{F}|/\delta)}{M_{\mathsf{sim}}}}.$$

# D  PROOFS IN SECTION 3

## D.1  PROOF OF THEOREM 1

We first define a specific IR decomposition for any function $f$ in Definition 2 that is useful in the rest of the proof.

**Lemma 3** (Standardized IR Decomposition). *For any function $f$ with interaction rank $K$ and training distribution $x \sim p, y_i \sim p_i(\cdot|x), \forall i$, there exists a group of sub-functions $\cup_{0 \le k \le K-1}\{g'_{j_1, \cdots, j_k}\}_{j_1 < \cdots < j_k}$ where*

$$\mathbb{E}_{y_{j_l} \sim p_{j_l}(\cdot|x)}[g'_{j_1, \cdots, j_k}(x, y_{j_1}, \cdots, y_{j_k})] = 0 \tag{5}$$

*for all $k \in [K-1], l \in [k], x \in \mathcal{X}, y_{j_{l'}} \in \mathcal{Y}_{j_{l'}}(l' \ne l)$ and*

$$f(x, y_1, \cdots, y_W) = \sum_{k=0}^{K-1} \sum_{1 \le j_1 < \cdots < j_k \le W} g'_{j_1, \cdots, j_k}(x, y_{j_1}, \cdots, y_{j_k}), \forall x \in \mathcal{X}, y_1 \in \mathcal{Y}_1, \cdots, y_W \in \mathcal{Y}_W.$$

*We call this group of sub-functions $\cup_{0 \le k \le K-1}\{g'_{j_1, \cdots, j_k}\}_{j_1 < \cdots < j_k}$ the standardized IR decomposition of $f$.*

The standardized decomposition separates the variations and mean of $f$ under the training distribution. With Lemma 3, we are able to provide an upper bound per-sub-function fitting error by simply fitting their summation $f$:

**Lemma 4** (Sub-function Alignment). *For any functions $f^\star$ and $\widehat{f}$ with interaction rank $K$, let $\cup_{0 \le k \le K-1}\{g_{j_1, \cdots, j_k}\}_{j_1 < \cdots < j_k}$ and $\cup_{0 \le k \le K-1}\{\widehat{g}_{j_1, \cdots, j_k}\}_{j_1 < \cdots < j_k}$ denote the standardized decomposition of $f^\star$ and $\widehat{f}$ in Lemma 3. Assume that the following holds*

$$\mathbb{E}_{x \sim p, y_1 \sim p_1(\cdot|x), \cdots, y_W \sim p_W(\cdot|x)}\left[\left((f^\star - \widehat{f})(x, y_1, \cdots, y_W)\right)^2\right] \le \epsilon.$$

*Then for any $0 \le k \le K-1$ and $1 \le j_1 < \cdots < j_k \le W$, we have:*

$$\mathbb{E}_{x \sim p, y_{j_1} \sim p_{j_1}(\cdot|x), \cdots, y_{j_k} \sim p_{j_k}(\cdot|x)}\left[(\Delta_{j_1, \cdots, j_k}(x, y_{j_1}, \cdots, y_{j_k}))^2\right] \le 2^k \epsilon,$$

*where $\Delta_{j_1, \cdots, j_k} := g_{j_1, \cdots, j_k} - \widehat{g}_{j_1, \cdots, j_k}$.*

Lemma 4 implies that the learning error of *standardized* sub-functions can be upper bounded by the fitting error of $f$ efficiently when the interaction rank is small. This property is the key reason why interaction rank is a more precise measure of the function complexity than the input size.

Now let $\cup_{0 \le k \le K-1}\{g_{j_1, \cdots, j_k}\}_{j_1 < \cdots < j_k}$ and $\cup_{0 \le k \le K-1}\{\widehat{g}_{j_1, \cdots, j_k}\}_{j_1 < \cdots < j_k}$ denote the standardized decomposition of $f^\star$ and $\widehat{f}$. Then from Lemma 4, we know for any $0 \le k \le K-1$ and $j_1 < \cdots < j_k$ that

$$\mathbb{E}_{x \sim p, y_{j_1} \sim p_{j_1}(\cdot|x), \cdots, y_{j_k} \sim p_{j_k}(\cdot|x)}\left[(\Delta_{j_1, \cdots, j_k}(x, y_{j_1}, \cdots, y_{j_k}))^2\right] \le 2^k \epsilon,$$

which implies that

$$\mathbb{E}_{x \sim p', y_{j_1} \sim p'_{j_1}(\cdot|x), \cdots, y_{j_k} \sim p'_{j_k}(\cdot|x)}\left[(\Delta_{j_1, \cdots, j_k}(x, y_{j_1}, \cdots, y_{j_k}))^2\right] \le (C_{\mathsf{DS}})^{k+1} 2^k \epsilon. \tag{6}$$

On the other hand, we know

$$\mathbb{E}_{x \sim p', y_1 \sim p'_1(\cdot|x), \cdots, y_W \sim p'_W(\cdot|x)}\left[\left((f^\star - \widehat{f})(x, y_1, \cdots, y_W)\right)^2\right]$$

$$= \mathbb{E}_{x \sim p', y_1 \sim p'_1(\cdot|x), \cdots, y_W \sim p'_W(\cdot|x)}\left[\left(\sum_{k=0}^{K-1} \sum_{1 \le j_1 < \cdots < j_k \le W} \Delta_{j_1, \cdots, j_k}(x, y_{j_1}, \cdots, y_{j_k})\right)^2\right]$$

$$\lesssim W^{K-1} \sum_{k=0}^{K-1} \sum_{1 \le j_1 < \cdots < j_k \le W} \mathbb{E}_{x \sim p', y_{j_1} \sim p'_{j_1}(\cdot|x), \cdots, y_{j_k} \sim p'_{j_k}(\cdot|x)}\left[(\Delta_{j_1, \cdots, j_k}(x, y_{j_1}, \cdots, y_{j_k}))^2\right],$$

where the last step is due to AM-GM inequality. Now substitute Eq. (6) into the above inequality, we have

$$\mathbb{E}_{x \sim p', y_1 \sim p_1'(\cdot|x), \cdots, y_W \sim p_W'(\cdot|x)} \left[ \left( (f^\star - \widehat{f})(x, y_1, \cdots, y_W) \right)^2 \right] \lesssim (2W)^{2(K-1)} C_{\mathsf{DS}}^K \epsilon.$$

This concludes our proof.

### D.2 PROOF OF LEMMA 3

From Definition 2, we know that there exists a group of sub-functions $\cup_{0 \le k \le K-1} \{g_{j_1, \cdots, j_k}\}_{j_1 < \cdots < j_k}$ which satisfies

$$f(x, y_1, \cdots, y_W) = \sum_{k=0}^{K-1} \sum_{1 \le j_1 < \cdots < j_k \le W} g_{j_1, \cdots, j_k}(x, y_{j_1}, \cdots, y_{j_k}), \forall x \in \mathcal{X}, y_1 \in \mathcal{Y}_1, \cdots, y_W \in \mathcal{Y}_W.$$

We prove the proposition with induction on $K$. First for $K = 1$, Lemma 3 holds naturally. Now we suppose the proposition holds for $K-1$ where $K \ge 2$. Then for any $\{j_l\}_{l=1}^{K-1}$, we can construct $g'_{j_1, \cdots, j_{K-1}}(x, y_{j_1}, \cdots, y_{j_{K-1}})$ as follows:

$$g'_{j_1, \cdots, j_{K-1}}(x, y_{j_1}, \cdots, y_{j_{K-1}}) = g_{j_1, \cdots, j_{K-1}}(x, y_{j_1}, \cdots, y_{j_{K-1}})$$
$$+ \sum_{k=1}^{K-1} (-1)^k \sum_{1 \le l_1 < \cdots < l_k \le K-1} \mathbb{E}_{y_{j_{l_1}} \sim p_{j_{l_1}}(\cdot|x), \cdots, y_{j_{l_k}} \sim p_{j_{l_k}}(x)} \left[ g_{j_1, \cdots, j_{K-1}}(x, y_{j_1}, \cdots, y_{j_{K-1}}) \right].$$

It can be verified that $g'_{j_1, \cdots, j_{K-1}}(x, y_{j_1}, \cdots, y_{j_{K-1}})$ satisfies the property of standardized decomposition, i.e., Eq. (5). Now consider the function $f'$:

$$f'(x, y_1, \cdots, y_W) = f(x, y_1, \cdots, y_W) - \sum_{1 \le j_1 < \cdots < j_{K-1} \le W} g'_{j_1, \cdots, j_{K-1}}(x, y_{j_1}, \cdots, y_{j_{K-1}}).$$

Note that $f'$ satisfies Definition 2 with IR $K-1$. By induction hypothesis, we know there exists a standardized decomposition for $f'$:

$$f'(x, y_1, \cdots, y_W) = \sum_{k=0}^{K-2} \sum_{1 \le j_1 < \cdots < j_k \le W} g'_{j_1, \cdots, j_k}(x, y_{j_1}, \cdots, y_{j_k}), \forall x \in \mathcal{X}, y_1 \in \mathcal{Y}_1, \cdots, y_W \in \mathcal{Y}_W.$$

where $g'_{j_1, \cdots, j_k}$ satisfies the requirement in Eq. (5) for all $k \in [K-2]$. This implies that we have

$$f(x, y_1, \cdots, y_W) = \sum_{k=0}^{K-1} \sum_{1 \le j_1 < \cdots < j_k \le W} g'_{j_1, \cdots, j_k}(x, y_{j_1}, \cdots, y_{j_k}), \forall x \in \mathcal{X}, y_1 \in \mathcal{Y}_1, \cdots, y_W \in \mathcal{Y}_W,$$

where $g'_{j_1, \cdots, j_k}$ satisfies the requirement in Eq. (5) for all $k \in [K-1]$. Therefore the argument holds for $K$ as well. By induction we can prove the proposition.

### D.3 PROOF OF LEMMA 4

Fix any $0 \le k \le K-1$ and $1 \le j_1 < \cdots < j_k \le W$. With slight abuse of notations, we also use $f^\star$ and $\widehat{f}$ to denote the expected function value under the training distribution:

$$f^\star(x, y_{j_1}, \cdots, y_{j_k}) := \mathbb{E}_{y_j \sim p_j(\cdot|x), \forall j \notin \{j_l\}_{l \in [k]}} \left[ f(x, y_1, \cdots, y_W) \right],$$
$$\widehat{f}(x, y_{j_1}, \cdots, y_{j_k}) := \mathbb{E}_{y_j \sim p_j(\cdot|x), \forall j \notin \{j_l\}_{l \in [k]}} \left[ \widehat{f}(x, y_1, \cdots, y_W) \right].$$

From Cauchy-Schwartz inequality, we can observe that

$$\mathbb{E}_{x \sim p, y_{j_l} \sim p_{j_l}(\cdot|x), \forall l \in [k]} \left[ \left( (f^\star - \widehat{f})(x, y_{j_1}, \cdots, y_{j_k}) \right)^2 \right] \le \epsilon. \tag{7}$$

Since we are considering standardized decomposition, from Lemma 3 we have

$$f^\star(x, y_{j_1}, \cdots, y_{j_k}) = \sum_{k'=0}^{k} \sum_{1 \le l_1 < \cdots < l_{k'} \le k} g_{j_{l_1}, \cdots, j_{l_{k'}}}(x, y_{j_{l_1}}, \cdots, y_{j_{l_{k'}}}),$$

$$\widehat{f}(x, y_{j_1}, \cdots, y_{j_k}) = \sum_{k'=0}^{k} \sum_{1 \le l_1 < \cdots < l_{k'} \le k} \widehat{g}_{j_{l_1}, \cdots, j_{l_{k'}}}(x, y_{j_{l_1}}, \cdots, y_{j_{l_{k'}}}).$$

Now we use symmetrization trick to prove the result. Consider the following symmetrization operation of function $f$:

$$G(f^\star)(x, \{y_{j_l}\}_{l \in [k]}, \{y'_{j_l}\}_{l \in [k]}) := \sum_{k'=0}^{k} (-1)^{k'} \sum_{1 \le l_1 < \cdots < l_{k'} \le k} f^\star(x, \{y_{j_l}\}_{l \notin \{l_1, \cdots, l_{k'}\}}, \{y'_{j_l}\}_{l \in \{l_1, \cdots, l_{k'}\}}).$$

It can be verified that

$$G(f^\star)(x, \{y_{j_l}\}_{l \in [k]}, \{y'_{j_l}\}_{l \in [k]}) = \sum_{k'=0}^{k} (-1)^{k'} \sum_{1 \le l_1 < \cdots < l_{k'} \le k} g_{j_1, \cdots, j_k}(x, \{y_{j_l}\}_{l \notin \{l_1, \cdots, l_{k'}\}}, \{y'_{j_l}\}_{l \in \{l_1, \cdots, l_{k'}\}}).$$

This implies that we have

$$(G(f^\star - \widehat{f}))(x, \{y_{j_l}\}_{l \in [k]}, \{y'_{j_l}\}_{l \in [k]}) = \sum_{k'=0}^{k} (-1)^{k'} \sum_{1 \le l_1 < \cdots < l_{k'} \le k} \Delta_{j_1, \cdots, j_k}(x, \{y_{j_l}\}_{l \notin \{l_1, \cdots, l_{k'}\}}, \{y'_{j_l}\}_{l \in \{l_1, \cdots, l_{k'}\}}).$$

(8)

On the one hand, from AM-GM inequality and Eq. (7) we have

$$\mathbb{E}_{x \sim p, y_{j_l} \sim p_{j_l}(\cdot|x), y'_{j_l} \sim p_{j_l}(\cdot|x), \forall l \in [k]} \left[ \left( (G(f^\star - \widehat{f}))(x, \{y_{j_l}\}_{l \in [k]}, \{y'_{j_l}\}_{l \in [k]}) \right)^2 \right] \le 2^{2k} \epsilon.$$

On the other hand, we can expand the left hand side of the above inequality:

$$\mathbb{E}_{x \sim p, y_{j_l} \sim p_{j_l}(\cdot|x), y'_{j_l} \sim p_{j_l}(\cdot|x), \forall l \in [k]} \left[ \left( (G(f^\star - \widehat{f}))(x, \{y_{j_l}\}_{l \in [k]}, \{y'_{j_l}\}_{l \in [k]}) \right)^2 \right]$$

$$= \mathbb{E}_{x \sim p, y_{j_l} \sim p_{j_l}(\cdot|x), y'_{j_l} \sim p_{j_l}(\cdot|x), \forall l \in [k]} \left[ \left( \sum_{k'=0}^{k} (-1)^{k'} \sum_{1 \le l_1 < \cdots < l_{k'} \le k} \Delta_{j_1, \cdots, j_k}(x, \{y_{j_l}\}_{l \notin \{l_1, \cdots, l_{k'}\}}, \{y'_{j_l}\}_{l \in \{l_1, \cdots, l_{k'}\}}) \right)^2 \right]$$

$$= 2^k \mathbb{E}_{x \sim p, y_{j_l} \sim p_{j_l}(\cdot|x), \forall l \in [k]} \left[ (\Delta_{j_1, \cdots, j_k}(x, y_{j_1}, \cdots, y_{j_k}))^2 \right],$$

where the second step is due to Eq. (8) and the third step is because the cross terms are 0 due to the independence between $y_j$ and $y'_j$ given $x$ and Lemma 3. Therefore we have

$$\mathbb{E}_{x \sim p, y_{j_1} \sim p_{j_1}(\cdot|x), \cdots, y_{j_k} \sim p_{j_k}(\cdot|x)} \left[ (\Delta_{j_1, \cdots, j_k}(x, y_{j_1}, \cdots, y_{j_k}))^2 \right] \le 2^k \epsilon,$$

which concludes our proof.

# E   PROOF OF THEOREM 2

We first present the formal statement of Theorem 2:

**Theorem 4.** *Suppose Assumption 1 hold. Let* $\Pi_i(C) := \{\mu_i : \mathbb{E}_{c\sim\rho}[\chi^2(\mu_i(c), \nu_i(c))] \leq C\}$ *denote the policy class which has bounded* $\chi^2$*-divergence from the behavior policy* $\nu_i$. *Fix any* $\delta \in (0, 1]$ *and select*

$$T = (2N^2)^{-\frac{2K-2}{3K-1}}\epsilon^{-\frac{2}{3K-1}}, \qquad \eta = \lambda = (2N^2)^{\frac{K-1}{3K-1}}\epsilon^{\frac{1}{3K-1}}.$$

*Then with probability at least* $1 - \delta$, *we have*

$$\max_i \mathsf{Gap}_i(\widehat{\pi}) \lesssim \max_{i\in[N]} \min_{C\geq 1} \left\{ C\left((2N^2)^{K-1}\epsilon\right)^{\frac{1}{3K-1}} + \mathsf{subopt}_i(C, \widehat{\pi}) \right\},$$

*where* $\mathsf{subopt}_i(C, \widehat{\pi}) := \max_{\mu_i\in\Pi_i} r_i^\star(\mu_i, \widehat{\pi}_{-i}) - \max_{\mu_i\in\Pi_i(C)} r_i^\star(\mu_i, \widehat{\pi}_{-i})$ *is the off-support bias.*

**Proof of Theorem 4.**   Note that for any agent $i \in [N]$ and policy $\mu_i \in \Pi_i(C)$ where $C > 1$, we have

$$\sum_{t=1}^T \left(r_i^\star(\mu_i, \pi_{-i}^t) - r_i^\star(\pi^t)\right) = \underbrace{\sum_{t=1}^T \mathbb{E}_{c\sim\rho, a_i\sim\mu_i(c), \boldsymbol{a}_{-i}\sim\pi_{-i}^t}[(r_i^\star - \widehat{r}_i)(c, \boldsymbol{a})]}_{(1)}$$

$$+ \underbrace{\sum_{t=1}^T \mathbb{E}_{c\sim\rho, \boldsymbol{a}\sim\pi^t}[(\widehat{r}_i - r_i^\star)(c, \boldsymbol{a})]}_{(2)} + \underbrace{\sum_{t=1}^T \left(\widehat{r}_i(\mu_i, \pi_{-i}^t) - \widehat{r}_i(\pi^t)\right)}_{(3)}. \tag{9}$$

With slight abuse of the notations, we use $\cup_{0\leq k\leq K-1}\{g_{j_1,\cdots,j_k}^i\}$ and $\cup_{0\leq k\leq K-1}\{\widehat{g}_{j_1,\cdots,j_k}^i\}$ to denote the standardized decomposition of $r_i^\star$ and $\widehat{r}_i$, as defined in Lemma 3. We also use $\Delta_{j_1,\cdots,j_k}^i$ to denote $g_{j_1,\cdots,j_k}^i - \widehat{g}_{j_1,\cdots,j_k}^i$. First note that from Lemma 4, we have for all $i \in [N], 0 \leq k \leq K-1$ and $1 \leq j_1 < \cdots < j_k \leq N$ where $j_l \neq i$ for all $l \in [k]$ that:

$$\mathbb{E}_{c\sim\rho, a_i\sim\nu_i(\cdot|c), a_{j_l}\sim\nu_{j_l}(\cdot|c), \forall l\in[k]} \left[\left(\Delta_{j_1,\cdots,j_k}^i(c, a_i, a_{j_1}, \cdots, a_{j_k})\right)^2\right] \leq 2^k\epsilon, \tag{10}$$

Next we bound terms (1), (2) and (3) in Eq. (9) respectively.

**Bounding term (1).**   For term (1), from Lemma 3, we know for all policy $\mu_i \in \Pi_i(C)$ where $C \geq 1$ that:

$$\mathbb{E}_{c\sim\rho, a_i\sim\mu_i(\cdot|c), \boldsymbol{a}_{-i}\sim\pi_{-i}^t(\cdot|c)}[(r_i^\star - \widehat{r}_i)(c, \boldsymbol{a})]$$

$$= \sum_{k=0}^K \sum_{1\leq j_1<\cdots<j_k\leq N: j_l\neq i, \forall l\in[k]} \mathbb{E}_{c\sim\rho, a_i\sim\mu_i(\cdot|c), a_{j_l}\sim\pi_{j_l}^t(\cdot|c), \forall l\in[k]} \left[\Delta_{j_1,\cdots,j_k}^i(c, a_i, a_{j_1}, \cdots, a_{j_k})\right].$$

To quantify the above transfer error, we have the following lemma which leverages the $\chi^2$-divergence between the target distribution and training distribution:

**Lemma 5.** *For two distributions* $d^1, d^2 \in \Delta(\mathcal{Z})$ *and any function* $f$ *defined on* $\mathcal{Z}$, *we have*

$$\mathbb{E}_{z\sim d^1}[f(z)] \leq \sqrt{\mathbb{E}_{z\sim d^2}[(f(z))^2](1 + \chi^2(d^1, d^2))}.$$

*Proof.*  Note that we have

$$1 + \chi^2(d^1, d^2) = 1 + \sum_{z\in\mathcal{Z}} \frac{\left(d^1(z) - d^2(z)\right)^2}{d^2(z)} = \sum_{z\in\mathcal{Z}} \frac{\left(d^1(z)\right)^2}{d^2(z)}.$$

Then the lemma comes directly from Cauchy-Schwartz inequality.  □

From Lemma 5 we have

$$\mathbb{E}_{c\sim\rho, a_i\sim\mu_i(\cdot|c), a_{j_l}\sim\pi_{j_l}^t(\cdot|c), \forall l\in[k]} \left[\Delta_{j_1,\cdots,j_k}^i(c, a_i, a_{j_1}, \cdots, a_{j_k})\right]$$

$$\leq \sqrt{\mathbb{E}_{c\sim\rho, a_i\sim\nu_i(c), a_{j_l}\sim\nu_{j_l}(\cdot|c), \forall l\in[k]} \left[\left(\Delta_{j_1,\cdots,j_k}^i(c, a_i, a_{j_1}, \cdots, a_{j_k})\right)^2\right]}$$

$$\cdot \sqrt{\left(\left(1 + \chi^2\left(\rho\circ\left(\mu_i\times\prod_{l\in[k]}\pi_{j_l}^t\right), \rho\circ\left(\nu_i\times\prod_{l\in[k]}\nu_{j_l}\right)\right)\right)\right)}$$

$$\leq \sqrt{2^k\epsilon\left(1 + \chi^2\left(\rho\circ\left(\mu_i\times\prod_{l\in[k]}\pi_{j_l}^t\right), \rho\circ\left(\nu_i\times\prod_{l\in[k]}\nu_{j_l}\right)\right)\right)},$$

where recall that we use $\rho\circ p$ to denote the joint distribution $c\sim\rho, a\sim p(\cdot|c)$ for some conditional distribution $p$. In the last step we utilize Eq. (10).

Now we only need to bound $\chi^2$-divergence between $\rho\circ\mu_i\circ\prod_{l\in[k]}\pi_{j_l}^t$ and $\rho\circ\nu_i\circ\prod_{l\in[k]}\nu_{j_l}$. We achieve this with the following lemma:

**Lemma 6.** *For any $2k$ policies $\{p_j\}_{j=1}^k$ and $\{q_j\}_{j=1}^k$, we have*

$$1 + \chi^2\left(\rho\circ\prod_{j=1}^k p_j, \rho\circ\prod_{j=1}^k q_j\right) = \mathbb{E}_{c\sim\rho}\left[\prod_{j=1}^k\left(1 + \chi^2\left(p_j(c), q_j(c)\right)\right)\right].$$

*Proof.* Note that we have

$$1 + \chi^2\left(\rho\circ\prod_{j=1}^k p_j, \rho\circ\prod_{j=1}^k q_j\right) = \sum_{c,a_1,\cdots,a_k} \frac{\left(\rho(c)\prod_{j\in[k]}p_j(a_j|c)\right)^2}{\rho(c)\prod_{j\in[k]}q_j(a_j|c)}$$

$$= \sum_c \rho(c) \sum_{a_1,\cdots,a_k} \frac{\left(\prod_{j\in[k]}p_j(a_j|c)\right)^2}{\prod_{j\in[k]}q_j(a_j|c)} = \sum_{c\in\mathcal{S}} \rho(c) \prod_{j\in[k]}\left(\sum_{a_j}\frac{(p_j(a_j|c))^2}{q_j(a_j|c)}\right)$$

$$= \sum_c \rho(c) \prod_{j\in[k]}\left(1 + \chi^2\left(p_j(c), q_j(c)\right)\right) = \mathbb{E}_{c\sim\rho}\left[\prod_{j=1}^k\left(1 + \chi^2\left(p_j(c), q_j(c)\right)\right)\right].$$

$\square$

Therefore, from Lemma 6 we have

$$1 + \chi^2\left(\rho\circ\left(\mu_i\times\prod_{l\in[k]}\pi_{j_l}^t\right), \rho\circ\left(\nu_i\times\prod_{l\in[k]}\nu_{j_l}\right)\right)$$

$$= \mathbb{E}_{c\sim\rho}\left[\left(\chi^2(\mu_i(c), \nu_i(c)) + 1\right)\prod_{l\in[k]}\left(\chi^2(\pi_{j_l}^t(c), \nu_{j_l}(c)) + 1\right)\right]. \tag{11}$$

Meanwhile, from the policy update formula Eq. (2), we have for all $t\in[T]$ and $c\in\mathcal{C}$:

$$-\langle\widehat{r}_i^t(c, \cdot), \pi_i^{t+1}(c)\rangle + \lambda\chi^2(\pi_i^{t+1}(c), \nu_i(c)) + \frac{1}{\eta}D_{c,i}(\pi_i^{t+1}(c), \pi_i^t(c))$$

$$\leq -\langle\widehat{r}_i^t(c, \cdot), \pi_i^t(c)\rangle + \lambda\chi^2(\pi_i^t(c), \nu_i(c)) + \frac{1}{\eta}D_{c,i}(\pi_i^t(c), \pi_i^t(c)).$$

Note that $D_{c,i}(\pi_i^t(c), \pi_i^t(c)) = 0$ and $\widehat{r}_i^t\in[0,1]$, we know

$$\chi^2(\pi_i^{t+1}(c), \nu_i(c)) \leq \chi^2(\pi_i^t(c), \nu_i(c)) + \frac{1}{\lambda}.$$

Since $\chi^2(\pi_i^1(c), \nu_i(c)) = \chi^2(\nu_i(c), \nu_i(c)) = 0$, for all $t \in [T]$ and $s \in \mathcal{S}$ we have

$$\chi^2(\pi_i^t(c), \nu_i(c)) \le \frac{t-1}{\lambda}, \forall t \in [T+1]. \tag{12}$$

Substitute Eq. (12) into Eq. (11) and we have

$$1 + \chi^2 \left( \rho \circ \left( \mu_i \times \prod_{l \in [k]} \pi_{j_l}^t \right), \rho \circ \left( \nu_i \times \prod_{l \in [k]} \nu_{j_l} \right) \right) \le \left( \frac{T}{\lambda} \right)^k \mathbb{E}_{c \sim \rho} \left[ (\chi^2(\mu_i(c), \nu_i(c)) + 1) \right] \le (C+1) \left( \frac{T}{\lambda} \right)^k,$$

where the second step is due to $\mu_i \in \Pi_i(C)$.

Therefore, we have for all policies $\mu_i \in \Pi_i(C)$ where $C \ge 1$ that

$$\mathbb{E}_{c \sim \rho, a_i \sim \mu_i(c), a_{j_l} \sim \pi_{j_l}^t(\cdot|c), \forall l \in [k]} \left[ \Delta_{j_1, \cdots, j_k}^i(c, a_i, a_{j_1}, \cdots, a_{j_k}) \right] \lesssim \sqrt{C\epsilon \cdot \left( \frac{2T}{\lambda} \right)^k}.$$

This implies that we have

$$(1) \lesssim T \sum_{k=0}^{K-1} \mathbb{C}_{N-1}^k \sqrt{C\epsilon_i \cdot \left( \frac{T}{\lambda} \right)^k} \lesssim T \sqrt{C\epsilon \cdot \left( \frac{2TN^2}{\lambda} \right)^{K-1}}. \tag{13}$$

Here $\mathbb{C}$ is the combination number.

**Bounding term (2).** Similarly, for term (2), following the same arguments as bounding term (1), we know for all policy $\mu_i \in \Pi_i(C)$ where $C \ge 1$ that:

$$\mathbb{E}_{c \sim \rho, a_i \sim \pi_i^t(c), a_{j_l} \sim \pi_{j_l}^t(c), \forall l \in [k]} \left[ \Delta_{j_1, \cdots, j_k}^i(c, a_i, a_{j_1}, \cdots, a_{j_k}) \right] \le \sqrt{\epsilon \left( \mathbb{E}_{x \sim \rho}[f_{c,i}(\pi_i^t)] + 1 \right) \cdot \left( \frac{2T}{\lambda} \right)^k}.$$

Recall that we use $f_{c,i}(p)$ to denote the chi-squared divergence $\chi^2(p, \nu_i(c))$. Then with AM-GM inequality, we have

$$\mathbb{E}_{c \sim \rho, a_i \sim \pi_i^t(c), a_{j_l} \sim \pi_{j_l}^t(c), \forall l \in [k]} \left[ \Delta_{j_1, \cdots, j_k}^i(c, a_i, a_{j_1}, \cdots, a_{j_k}) \right]$$

$$\le \frac{\lambda}{N^{K-1}} \mathbb{E}_{x \sim \rho}[f_{c,i}(\pi_i^t)] + \frac{N^{K-1}}{\lambda} \cdot \left( \frac{2T}{\lambda} \right)^k \cdot \epsilon + \sqrt{\epsilon \cdot \left( \frac{2T}{\lambda} \right)^k}.$$

Therefore, we have

$$(2) - \lambda \sum_{t=1}^T \mathbb{E}_{x \sim \rho}[f_{c,i}(\pi_i^t)] \lesssim \frac{T}{\lambda} \cdot \left( \frac{2TN^2}{\lambda} \right)^{K-1} \cdot \epsilon + T \sqrt{\epsilon \cdot \left( \frac{2TN^2}{\lambda} \right)^{K-1}}. \tag{14}$$

**Bounding term (3).** First we have the following lemma to characterize the no-regret guarantee of regularized policy gradient (see Appendix E.1 for proof):

**Lemma 7** (No-Regret Regularized Policy Gradient). *Given a sequence of loss functions $\{l^t\}_{t \in [T]}$ where $l^t : \mathcal{X} \times \mathcal{Y} \to [0, B]$ for some $B > 0$ and a reference policy $\nu : \mathcal{X} \mapsto \Delta_{\mathcal{Y}}$. Suppose we initialize $p^1$ to be $\nu$ and run the following regularized policy gradient for $T$ iterations:*

$$p^{t+1}(x) = \arg \min_{p \in \Delta_{\mathcal{Y}}} -\langle l^t(x, \cdot), p \rangle + \lambda \chi^2(p, \nu(x)) + \frac{1}{\eta} D_x(p, p^t),$$

*where $D_x(p, p^t)$ is the Bregman divergence between $p(x)$ and $p^t(x)$. Then we have for all policy $\mu$ and $x \in \mathcal{X}$ that*

$$\sum_{t=1}^T \langle l^t(x), \mu(x) - p^t(x) \rangle + \lambda \sum_{t=1}^{T+1} \chi^2(p^t(x), \nu(x)) \le \left( T\lambda + \frac{1}{\eta} \right) \chi^2(\mu(x), \nu(x)) + \frac{\eta T B^2}{4}.$$

Note that $(3) = \sum_{t=1}^T \mathbb{E}_{x \sim \rho}[\langle \widehat{r}_i^t(c), \mu(c) - \pi^t(c) \rangle]$. Thus, Lemma 7 implies that for any policy $\mu_i \in \Pi_i(C)$, we have:

$$(3) + \lambda \sum_{t=1}^T \mathbb{E}_{x \sim \rho}[f_{c,i}(\pi_i^t)] \lesssim TC\lambda + \frac{C}{\eta} + \frac{\eta T}{4}. \tag{15}$$

**Putting all pieces together.** Now substituting Eq (13),(14),(15) into Eq (9), we have for all policy $\mu_i \in \Pi_i(C)$ where $C \geq 1$ that

$$r_i^\star(\mu_i, \widehat{\pi}_{-i}) - r_i^\star(\widehat{\pi}) \lesssim C\lambda + \frac{C}{\eta T} + \frac{\eta}{4} + \sqrt{C\epsilon \cdot \left(\frac{2TN^2}{\lambda}\right)^{K-1}} + \frac{1}{\lambda} \cdot \left(\frac{2TN^2}{\lambda}\right)^{K-1} \cdot \epsilon.$$

Therefore by setting

$$T = (2N^2)^{-\frac{2K-2}{3K-1}} \epsilon^{-\frac{2}{3K-1}}, \qquad \eta = \lambda = (2N^2)^{\frac{K-1}{3K-1}} \epsilon^{\frac{1}{3K-1}},$$

we have for all policy $\mu_i \in \Pi_i(C)$ where $C \geq 1$ that

$$r_i^\star(\mu_i, \widehat{\pi}_{-i}) - r_i^\star(\widehat{\pi}) \lesssim C\left((2N^2)^{K-1}\epsilon\right)^{\frac{1}{3K-1}}.$$

This concludes our proof.

### E.1   PROOF OF LEMMA 7

Let $f_x(p)$ denote the $\chi^2$-divergence $\chi^2(p(x), \nu(x))$. First due to first order optimality in the policy update step , we know for all $p : \mathcal{X} \to \Delta_{\mathcal{Y}}$ and all $t \in [T], x \in \mathcal{X}$ that:

$$\left\langle -\eta l^t(x) + (1 + \eta\lambda)\nabla f_x(p^{t+1}) - \nabla f_x(p^t), p(x) - p^{t+1}(x) \right\rangle \geq 0. \tag{16}$$

This implies that for all $t \in [T], x \in \mathcal{X}$ and any policy $\mu$, we have

$$
\begin{aligned}
&\left\langle \eta l^t(x), \mu(x) - p^t(x) \right\rangle + \eta\lambda f_x(p^t) - \eta\lambda f_x(\mu) \\
={}& \left\langle \eta l^t(x) - (1 + \eta\lambda)\nabla f_x(p^{t+1}) + \nabla f_x(p^t), \mu(x) - p^{t+1}(x) \right\rangle \\
&+ \left\langle \nabla f_x(p^{t+1}) - \nabla f_x(p^t), \mu(x) - p^{t+1}(x) \right\rangle + \left\langle \eta l^t(x), p^{t+1}(x) - p^t(x) \right\rangle \\
&+ \left\langle \eta\lambda \nabla f_x(p^{t+1}), \mu(x) - p^{t+1}(x) \right\rangle + \eta\lambda f_x(p^t) - \eta\lambda f_x(\mu), \\
\leq{}& \underbrace{\left\langle \nabla f_x(p^{t+1}) - \nabla f_x(p^t), \mu(x) - p^{t+1}(x) \right\rangle}_{(4)} + \underbrace{\left\langle \eta l^t(x), p^{t+1}(x) - p^t(x) \right\rangle}_{(5)} \\
&+ \underbrace{\left\langle \eta\lambda \nabla f_x(p^{t+1}), \mu(x) - p^{t+1}(x) \right\rangle + \eta\lambda f_x(p^t) - \eta\lambda f_x(\mu)}_{(6)}.
\end{aligned}
$$

Next we bound terms (4), (5) and (6) respectively.

First for term (4), note that we have the following lemma:

**Lemma 8.** *For any $i \in [N]$ and $p_1, p_2, p_3 : \mathcal{X} \to \Delta_{\mathcal{Y}}$, we have for all $x \in \mathcal{X}$*

$$\langle \nabla f_x(p_1) - \nabla f_x(p_2), p_3(x) - p_1(x) \rangle = D_x(p_3, p_2) - D_x(p_3, p_1) - D_x(p_1, p_2).$$

*Proof.* By definition, we know

$$D_x(p, p') = f_x(p) - f_x(p') - \langle \nabla f_x(p'), p - p' \rangle.$$

Substitute the definition into Lemma 8 and we can prove the lemma. $\qquad\square$

From Lemma 8, we can rewrite (4) as follows:

$$(4) = D_x(\mu, p^t) - D_x(\mu, p^{t+1}) - D_x(p^{t+1}, p^t).$$

Then for term (5), from Cauchy-Schwartz inequality, we have

$$(5) \leq \sum_{y \in \mathcal{Y}} \frac{(p^{t+1}(y|x) - p^t(y|x))^2}{\nu(y|x)} + \frac{\nu(y|x)\eta^2(l^t(x,y))^2}{4} \leq D_x(p^{t+1}, p^t) + \frac{\eta^2 B^2}{4},$$

where the last step comes from the definition of $D_x$.

Finally for term (6), Since $f_x$ is convex, we know

$$\langle \eta\lambda\nabla f_x(p^{t+1}), \mu(x) - p^{t+1}(x)\rangle \le \eta\lambda f_x(\mu) - \eta\lambda f_x(p^{t+1}).$$

This implies that

$$(6) \le \eta\lambda\left(f_x(p^t) - f_x(p^{t+1})\right).$$

In summary, for all $t \in [T], s \in \mathcal{S}$ and any policy $\mu$, we have

$$\langle \eta l^t(x), \mu(x) - p^t(x)\rangle + \eta\lambda f_x(p^t) - \eta\lambda f_x(\mu)$$

$$\le \left(D_x(\mu, p^t) - D_x(\mu, p^{t+1})\right) + \eta\lambda\left(f_x(p^t) - f_x(p^{t+1})\right) + \frac{\eta^2 B^2}{4}.$$

Therefore, summing up from $t = 1$ to $T$, we have

$$\sum_{t=1}^{T}\langle l^t(x), \mu(x) - p^t(x)\rangle + \lambda\sum_{t=1}^{T+1}\chi^2(p^t(x), \nu(x)) \le \left(T\lambda + \frac{1}{\eta}\right)\chi^2(\mu(x), \nu(x)) + \frac{\eta T B^2}{4},$$

where we use the fact that $D_x(\mu, p^1) = D_x(\mu, \nu) = \chi^2(p^t(x), \nu(x))$.

# F  PROOF OF THEOREM 3

Let $f_{c,s,i,h}(p)$ to denote the $\chi^2$-divergence $\chi^2(p_h(c,s), \nu_{i,h}(c,s))$. Note that for any agent $i \in [N]$ and policy $\mu_i \in \Pi_i(C)$ where $C \geq 1$, we have

$$
\sum_{t=1}^{T} \mathbb{E}_{c\sim\rho} \left[ V_{i,1}^{\mu_i \circ \pi_{-i}^t, r^\star}(c, \boldsymbol{s}_1) \right] - \mathbb{E}_{c\sim\rho} \left[ V_{i,1}^{\pi^t, r^\star}(c, \boldsymbol{s}_1) \right]
$$

$$
= \underbrace{\left( \sum_{h=1}^{H} \sum_{t=1}^{T} \mathbb{E}_{c\sim\rho, \boldsymbol{s}_h \sim d_h^{\mu_i \circ \pi_{-i}^t}(\cdot|c), a_{i,h} \sim \mu_{i,h}(\cdot|c,s_{i,h}), \boldsymbol{a}_{-i,h} \sim \pi_{-i}^t(\cdot|c,\boldsymbol{s}_{-i,h})} \left[ r_{i,h}^\star(c, \boldsymbol{s}_h, \boldsymbol{a}_h) - \widehat{r}_{i,h}(c, \boldsymbol{s}_h, \boldsymbol{a}_h) \right] \right)}_{(1)}
$$

$$
+ \underbrace{\left( \sum_{h=1}^{H} \sum_{t=1}^{T} \mathbb{E}_{c\sim\rho, \boldsymbol{s}_h \sim d_h^{\pi^t}(\cdot|c), \boldsymbol{a}_h \sim \pi^t(\cdot|c,\boldsymbol{s}_h)} \left[ -r_{i,h}^\star(c, \boldsymbol{s}_h, \boldsymbol{a}_h) + \widehat{r}_{i,h}(c, \boldsymbol{s}_h, \boldsymbol{a}_h) \right] \right)}_{(2)}
$$

$$
+ \underbrace{\left( \sum_{t=1}^{T} \mathbb{E}_{c\sim\rho} \left[ V_{i,1}^{\mu_i \circ \pi_{-i}^t, \widehat{r}}(c, \boldsymbol{s}_1) \right] - \mathbb{E}_{c\sim\rho} \left[ \widehat{V}_{i,1}^{\mu_i \circ \pi_{-i}^t, \widehat{r}}(c, \boldsymbol{s}_1) \right] \right)}_{(3)}
$$

$$
+ \underbrace{\left( \sum_{t=1}^{T} \mathbb{E}_{c\sim\rho} \left[ \widehat{V}_{i,1}^{\pi^t, \widehat{r}}(c, \boldsymbol{s}_1) \right] - \mathbb{E}_{c\sim\rho} \left[ V_{i,1}^{\pi^t, \widehat{r}}(c, \boldsymbol{s}_1) \right] \right)}_{(4)}
$$

$$
+ \underbrace{\left( \sum_{t=1}^{T} \mathbb{E}_{c\sim\rho} \left[ \widehat{V}_{i,1}^{\mu_i \circ \pi_{-i}^t, \widehat{r}}(c, \boldsymbol{s}_1) \right] - \mathbb{E}_{c\sim\rho} \left[ \widehat{V}_{i,1}^{\pi^t, \widehat{r}}(c, \boldsymbol{s}_1) \right] \right)}_{(5)},
$$

where we use $\widehat{V}_{i,h}^{\pi,\widehat{r}}$ to denote the joint value function under reward $\widehat{r}$ and transition $\widehat{P}$. Next we will bounded these terms separately. In particular, terms (1) and (2) are bounded by statistical guarantees on the reward model and the distribution shift robustness of low IR models; term (3) and (4) are bounded by the statistical guarantees of the transition model, while using the decoupling property, and term (5) is bounded by no-regret analysis while identifying proper value and Q functions that satisfies Bellman equation.

We use $\cup_{0 \leq k \leq K-1} \{g_{j_1,\cdots,j_k}^{i,h}\}$ and $\cup_{0 \leq k \leq K-1} \{\widehat{g}_{j_1,\cdots,j_k}^{i,h}\}$ to denote the standardized decomposition of $r_{i,h}^\star$ and $\widehat{r}_{i,h}$, as defined in Lemma 3. We also use $\Delta_{j_1,\cdots,j_k}^{i,h}$ to denote $g_{j_1,\cdots,j_k}^{i,h} - \widehat{g}_{j_1,\cdots,j_k}^{i,h}$. From Assumption 2 and the LSR guarantee Lemma 13, with probability at least $1 - \delta/2$ we have for all $i \in [N], h \in [H]$ that:

$$
\mathbb{E}_{c\sim\rho, s_j \sim \sigma_{j,h}(\cdot|c), a_j \sim \nu_{j,h}(\cdot|c,s_j), \forall j} \left[ \left( r_{h,i}^\star(c, \boldsymbol{s}, \boldsymbol{a}) - \widehat{r}_{h,i}(c, \boldsymbol{s}, \boldsymbol{a}) \right)^2 \right] \lesssim \frac{\log(NH|\mathcal{R}|/\delta)}{M} := \epsilon_{\mathsf{R}}.
$$

Combining the above inequality with Lemma 4, we have for all $i \in [N], h \in [H], 0 \leq k \leq K - 1$ and $1 \leq j_1 < \cdots < j_k \leq N$ where $j_l \neq i$ for all $l \in [k]$ that:

$$
\mathbb{E}_{c\sim\rho, s_i \sim \sigma_{i,h}(\cdot|c), a_i \sim \nu_i(\cdot|c,s_i), s_{j_l} \sim \sigma_{i,h}(\cdot|c), a_{j_l} \sim \nu_{j_l}(\cdot|c,s_{j_l}), \forall l \in [k]} \left[ \left( \Delta_{j_1,\cdots,j_k}^{i,h}(c, z_i, z_{j_1}, \cdots, z_{j_k}) \right)^2 \right] \leq 2^k \epsilon_{\mathsf{R}},
$$

where we use $z_j$ to denote $(s_j, a_j)$. Next we bound terms (1), (2) and (3) in Eq. (9) respectively.

For term (1), fix $h \in [H]$ and $t \in [T]$, then we know

$$
\mathbb{E}_{c\sim\rho, \boldsymbol{s}_h \sim d_h^{\mu_i \circ \pi_{-i}^t}(\cdot|c), a_{i,h} \sim \mu_{i,h}(c,s_{i,h}), \boldsymbol{a}_{-i,h} \sim \pi_{-i}^t(c,\boldsymbol{s}_{-i,h})} \left[ r_{i,h}^\star(c, \boldsymbol{s}_h, \boldsymbol{a}_h) - \widehat{r}_{i,h}(c, \boldsymbol{s}_h, \boldsymbol{a}_h) \right]
$$

$$
= \sum_{k=0}^{K-1} \sum_{1 \leq j_1 < \cdots < j_k \leq N: j_l \neq i, \forall l \in [k]} \mathbb{E}_{c\sim\rho, z_i \sim d_h^{\mu_i}(\cdot|c), z_{j_l} \sim d_h^{\pi_{j_l}^t}(\cdot|c), \forall l} [\Delta_{j_1,\cdots,j_k}^{i,h}(c, z_i, z_{j_1}, \cdots, z_{j_k})]
$$

With similar arguments in the proof of Theorem 2, from Lemma 5 we have

$$
\mathbb{E}_{c\sim\rho, z_i\sim d_h^{\mu_i}(\cdot|c), z_{j_l}\sim d_h^{\pi_{j_l}^t}(\cdot|c), \forall l}[\Delta_{j_1,\cdots,j_k}^{i,h}(c, z_i, z_{j_1}, \cdots, z_{j_k})]
$$

$$
\leq \sqrt{\mathbb{E}_{c\sim\rho, s_i\sim d_h^{\mu_i}(\cdot|c), a_i\sim\nu_{i,h}(\cdot|c,s_i), s_{j_l}\sim d_h^{\pi_{j_l}^t}(\cdot|c), a_{j_l}\sim\nu_{j_l,h}(\cdot|c,s_{j_l})\forall l}\left[\left(\Delta_{j_1,\cdots,j_k}^{i,h}(c, z_i, z_{j_1}, \cdots, z_{j_k})\right)^2\right]}
$$

$$
\cdot \sqrt{\left(1+\chi^2\left(\rho\circ\left(d_h^{\mu_i}\times\prod_{l\in[K]}d_h^{\pi_{j_l}^t}\right)\circ\left(\mu_i\times\prod_{l\in[k]}\pi_{j_l}^t\right), \rho\circ\left(d_h^{\mu_i}\times\prod_{l\in[K]}d_h^{\pi_{j_l}^t}\right)\circ\left(\nu_i\times\prod_{l\in[k]}\nu_{j_l}\right)\right)\right)}
$$

$$
\leq \sqrt{(C_{\mathsf{S}})^{k+1}2^k\epsilon_{\mathsf{R}}}
$$

$$
\cdot\sqrt{\left(1+\chi^2\left(\rho\circ\left(d_h^{\mu_i}\times\prod_{l\in[K]}d_h^{\pi_{j_l}^t}\right)\circ\left(\mu_i\times\prod_{l\in[k]}\pi_{j_l}^t\right), \rho\circ\left(d_h^{\mu_i}\times\prod_{l\in[K]}d_h^{\pi_{j_l}^t}\right)\circ\left(\nu_i\times\prod_{l\in[k]}\nu_{j_l}\right)\right)\right)}.
$$

On the other hand, from Lemma 6 we know

$$
1+\chi^2\left(\rho\circ\left(d_h^{\mu_i}\times\prod_{l\in[K]}d_h^{\pi_{j_l}^t}\right)\circ\left(\mu_i\times\prod_{l\in[k]}\pi_{j_l}^t\right), \rho\circ\left(d_h^{\mu_i}\times\prod_{l\in[K]}d_h^{\pi_{j_l}^t}\right)\circ\left(\nu_i\times\prod_{l\in[k]}\nu_{j_l}\right)\right) \lesssim C\left(\frac{TH}{\lambda}\right)^k.
$$

This implies that

$$
\mathbb{E}_{c\sim\rho, z_i\sim d_h^{\mu_i}(\cdot|c), z_{j_l}\sim d_h^{\pi_{j_l}^t}(\cdot|c), \forall l}[\Delta_{j_1,\cdots,j_k}^{i,h}(c, z_i, z_{j_1}, \cdots, z_{j_k})] \lesssim \sqrt{C_{\mathsf{S}}^{k+1}C\left(\frac{2TH}{\lambda}\right)^k\epsilon_{\mathsf{R}}}
$$

Therefore we have

$$
(1) \lesssim TH\sqrt{CC_{\mathsf{S}}^K\left(\frac{2THN^2}{\lambda}\right)^{K-1}\epsilon_{\mathsf{R}}}.
$$

Similarly, term (2) is bounded by

$$
(2) \lesssim TH\sqrt{\left(\frac{C_{\mathsf{S}}TH}{\lambda}\right)^K(2N^2)^{K-1}\epsilon_{\mathsf{R}}}.
$$

For term (3), note that we have

$$
(3) = \sum_{h=1}^{H}\sum_{t=1}^{T}\mathbb{E}_{c\sim\rho, \boldsymbol{s}_h\sim d_h^{\mu_i\circ\pi_{-i}^t}(\cdot|c), a_{i,h}\sim\mu_{i,h}(\cdot|c,s_{i,h}), \boldsymbol{a}_{-i,h}\sim\pi_{-i}^t(\cdot|c,\boldsymbol{s}_{-i,h})}[\widehat{r}_{i,h}(c, \boldsymbol{s}_h, \boldsymbol{a}_h)]
$$

$$
- \mathbb{E}_{c\sim\rho, \boldsymbol{s}_h\sim\widehat{d}_h^{\mu_i\circ\pi_{-i}^t}(\cdot|c), a_{i,h}\sim\mu_{i,h}(\cdot|c,s_{i,h}), \boldsymbol{a}_{-i,h}\sim\pi_{-i}^t(\cdot|c,\boldsymbol{s}_{-i,h})}[\widehat{r}_{i,h}(c, \boldsymbol{s}_h, \boldsymbol{a}_h)]
$$

$$
\leq \sum_{h=1}^{H}\sum_{t=1}^{T}\mathbb{E}_{c\sim\rho}\left[\sum_{\boldsymbol{s},\boldsymbol{a}}\left|d_h^{\mu_i\circ\pi_{-i}^t}(\boldsymbol{s},\boldsymbol{a}|c) - \widehat{d}_h^{\mu_i\circ\pi_{-i}^t}(\boldsymbol{s},\boldsymbol{a}|c)\right|\right].
$$

At the same time, due to decoupled transition, we have the following lemma:

**Lemma 9.** *For any policy product $\pi$, we have for all $h\in[H]$ that*

$$
\mathbb{E}_{c\sim\rho}\left[\sum_{\boldsymbol{s},\boldsymbol{a}}\left|d_h^{\pi}(\boldsymbol{s},\boldsymbol{a}|c) - \widehat{d}_h^{\pi}(\boldsymbol{s},\boldsymbol{a}|c)\right|\right] \leq \sum_{j=1}^{N}\mathbb{E}_{c\sim\rho}\left[\sum_{s_j,a_j}\left|d_h^{\pi_j}(s_j,a_j|c) - \widehat{d}_h^{\pi_j}(s_j,a_j|c)\right|\right]
$$

Thus, from Lemma 9, we only need to bound $\mathbb{E}_{c\sim\rho}\left[\sum_{s_j,a_j}\left|d_h^{\pi_j}(s_j,a_j|c) - \widehat{d}_h^{\pi_j}(s_j,a_j|c)\right|\right]$ for any agent $j$ and single-agent policy $\pi_j$. This is achieved in the following lemma:

**Lemma 10.** *For any $j \in [N]$ and single-agent policy $\pi_j$, we have for all $h \in [H]$ that*

$$\mathbb{E}_{c\sim\rho}\left[\sum_{s_j,a_j}\left|d_h^{\pi_j}(s_j,a_j|c) - \widehat{d}_h^{\pi_j}(s_j,a_j|c)\right|\right]$$

$$\leq \sum_{h'=1}^{h-1}\mathbb{E}_{c\sim\rho,(s_j,a_j)\sim d_{h'}^{\pi_j}(\cdot|c)}\left[\left\|\widehat{P}_{j,h'}(\cdot|c,s_j,a_j) - P_{j,h'}^{\star}(\cdot|c,s_j,a_j)\right\|_1\right].$$

On the other hand, from the guarantee of MLE in the literature (Liu et al., 2022; Zhan et al., 2022; 2023b) (Lemma 14), we know with probability at least $1-\delta/2$ that for all $j \in [N], h \in [H]$

$$\mathbb{E}_{c\sim\rho,s_j\sim\sigma_{j,h}(\cdot|c),a_j\sim\nu_{j,h}(\cdot|c,s_j)}\left[\left\|\widehat{P}_{j,h}(\cdot|c,s_j,a_j) - P_{j,h}^{\star}(\cdot|c,s_j,a_j)\right\|_1^2\right] \lesssim \frac{\log(HN|\mathcal{P}|/\delta)}{M} := \epsilon_{\mathsf{P}}. \tag{17}$$

From Lemma 5, this implies that with probability at least $1-\delta/2$, we have for all $j \in [N], h \in [H], t \in [T], \mu_i \in \Pi_i(C)$ that

$$\mathbb{E}_{c\sim\rho,(s_i,a_i)\sim d_h^{\mu_i}(\cdot|c)}\left[\left\|\widehat{P}_{i,h}(\cdot|c,s_i,a_i) - P_{i,h}^{\star}(\cdot|c,s_i,a_i)\right\|_1\right] \lesssim \sqrt{C_{\mathsf{S}}C\epsilon_{\mathsf{P}}}, \tag{18}$$

$$\mathbb{E}_{c\sim\rho,(s_j,a_j)\sim d_h^{\pi_j^t}(\cdot|c)}\left[\left\|\widehat{P}_{j,h}(\cdot|c,s_j,a_j) - P_{j,h}^{\star}(\cdot|c,s_j,a_j)\right\|_1\right] \lesssim \sqrt{\frac{C_{\mathsf{S}}TH\epsilon_{\mathsf{P}}}{\lambda}}.$$

Therefore, we have

$$(3) \lesssim H^2 T\sqrt{C_{\mathsf{S}}C\epsilon_{\mathsf{P}}} + H^2 TN\sqrt{\frac{C_{\mathsf{S}}TH\epsilon_{\mathsf{P}}}{\lambda}}.$$

For term (4), following the same arguments for term (3), we have

$$(4) \lesssim H^2 TN\sqrt{\frac{C_{\mathsf{S}}TH\epsilon_{\mathsf{P}}}{\lambda}}.$$

For term (5), we first need to show that the expected single-agent Q function $\widehat{Q}_{i,h}^t$ satisfies Bellman equation. In particular, let $\widehat{Q}^{\pi,\widehat{r}} := \mathbb{E}_{(s_j,a_j)\sim\widehat{d}_h^{\pi_j}(\cdot|c),\forall j\neq i}\left[\widehat{Q}_{i,h}^{\pi,\widehat{r}}(c,\boldsymbol{s},\boldsymbol{a})\right]$ for any product policy $\pi$ and we have the following lemma:

**Lemma 11.** *Given a joint policy $\pi_{-i}$ for agents except $i$, for all $i \in [N], h \in [H], c \in \mathcal{C}, s_i \in \mathcal{S}_i, a \in \mathcal{A}_i$ and policy $\mu_i$, we have*

$$\widehat{V}_{i,h}^{\mu_i\circ\pi_{-i},\widehat{r}}(c,s_i) := \mathbb{E}_{a_i\sim\mu_{i,h}(\cdot|c,s_i)}\left[\widehat{Q}_{i,h}^{\mu_i\circ\pi_{-i},\widehat{r}}(c,s_i,a_i)\right],$$

$$\widehat{Q}_{i,h}^{\mu_i\circ\pi_{-i},\widehat{r}}(c,s_i,a_i) = \mathbb{E}_{(\boldsymbol{s}_{-i},\boldsymbol{a}_{-i})\sim\widehat{d}_h^{\pi_{-i}}(\cdot|c)}\left[\widehat{r}_{i,h}(c,\boldsymbol{s},\boldsymbol{a})\right] + \mathbb{E}_{s_i'\sim\widehat{P}_{i,h}(\cdot|c,s_i,a_i)}\left[\widehat{V}_{i,h+1}^{\mu_i\circ\pi_{-i},\widehat{r}}(c,s_i')\right].$$

Lemma 11 indeed implies that $\widehat{Q}_{i,h}^{\mu_i\circ\pi_{-i},\widehat{r}}(c,s_i,a_i)$ is a *valid* Q function w.r.t. to the reward function $\mathbb{E}_{(\boldsymbol{s}_{-i},\boldsymbol{a}_{-i})\sim\widehat{d}_h^{\pi_{-i}}(\cdot|c)}\left[\widehat{r}_{i,h}(c,\boldsymbol{s},\boldsymbol{a})\right]$ under transition model $\widehat{P}$ and thus we have the following performance difference lemma:

**Lemma 12.** *Given a joint policy $\pi_{-i}$ for agents except $i$, for any policies $\mu_i$ and $\mu_i'$, we have*

$$\widehat{V}_{i,1}^{\mu_i'\circ\pi_{-i},\widehat{r}}(c,s_{i,1}) - \widehat{V}_{i,1}^{\mu_i\circ\pi_{-i},\widehat{r}}(c,s_{i,1}) = \sum_{h=1}^{H}\mathbb{E}_{s_{i,h}\sim\widehat{d}_h^{\mu_i'}(\cdot|c)}\left[\left\langle\widehat{Q}_{i,h}^{\mu_i\circ\pi_{-i},r}(c,s_{i,h},\cdot),\mu_{i,h}'(\cdot|c,s_{i,h}) - \mu_{i,h}(\cdot|c,s_{i,h})\right\rangle\right].$$

Now given Lemma 12, we have

$$(5) = \sum_{t=1}^{T}\mathbb{E}_{c\sim\rho}\left[\widehat{V}_{i,1}^{\mu_i\circ\pi_{-i}^t,\widehat{r}}(c,s_{i,1})\right] - \mathbb{E}_{c\sim\rho}\left[\widehat{V}_{i,1}^{\pi^t,\widehat{r}}(c,s_{i,1})\right]$$

$$= \sum_{h=1}^{H} \mathbb{E}_{c \sim \rho, s_i \sim \widehat{d}_h^{\mu_i}(\cdot|c)} \left[ \sum_{t=1}^{T} \left\langle \widehat{Q}_{i,h}^t(c, s_i, \cdot), \mu_{i,h}(\cdot|c, s_i) - \pi_{i,h}^t(\cdot|c, s_i) \right\rangle \right]$$

$$\leq \underbrace{\sum_{h=1}^{H} \mathbb{E}_{c \sim \rho, s_i \sim d_h^{\mu_i}(\cdot|c)} \left[ \sum_{t=1}^{T} \left\langle \widehat{Q}_{i,h}^t(c, s_i, \cdot), \mu_{i,h}(\cdot|c, s_i) - \pi_{i,h}^t(\cdot|c, s_i) \right\rangle \right]}_{(6)}$$

$$+ \underbrace{TH \sum_{h=1}^{H} \mathbb{E}_{c \sim \rho} \left[ \sum_{s_i} \left| \widehat{d}_h^{\mu_i}(s_i|c) - d_h^{\mu_i}(s_i|c) \right| \right]}_{(7)}$$

Apply Lemma 7 and since $\mu_i \in \Pi_i(C)$, we have

$$(6) \lesssim TH\lambda C + \frac{HC}{\eta} + \frac{\eta H^3 T}{4}.$$

From Lemma 10 and Eq. (18), we have

$$(7) \lesssim TH^3 \sqrt{C_{\mathsf{S}} C \epsilon_{\mathsf{P}}}.$$

Therefore, we have

$$\mathbb{E}_{c \sim \rho} \left[ V_{i,1}^{\mu_i \circ \widehat{\pi}_{-i}, r^\star}(c, \boldsymbol{s}_1) \right] - \mathbb{E}_{c \sim \rho} \left[ V_{i,1}^{\widehat{\pi}, r^\star}(c, \boldsymbol{s}_1) \right]$$

$$\lesssim H \sqrt{C \left( \frac{C_{\mathsf{S}} TH}{\lambda} \right)^K (2N^2)^{K-1} \epsilon_{\mathsf{R}}} + H\lambda C + \frac{HC}{T\eta} + \frac{\eta H^3}{4} + H^3 \sqrt{C_{\mathsf{S}} C \epsilon_{\mathsf{P}}} + H^2 N \sqrt{\frac{C_{\mathsf{S}} TH \epsilon_{\mathsf{P}}}{\lambda}}.$$

Let

$$T = C_{\mathsf{S}}^{-\frac{2K}{3K+2}} H^{\frac{4}{3K+2}} (2N^2)^{-\frac{2K-2}{3K+2}} \epsilon_{\mathsf{RP}}^{-\frac{2}{3K+2}}, \eta = C_{\mathsf{S}}^{\frac{K}{3K+2}} H^{-\frac{3K+4}{3K+2}} (2N^2)^{\frac{K-1}{3K+2}} \epsilon_{\mathsf{RP}}^{\frac{1}{3K+2}}, \lambda = C_{\mathsf{S}}^{\frac{K}{3K+2}} H^{\frac{3K}{3K+2}} (2N^2)^{\frac{K-1}{3K+2}} \epsilon_{\mathsf{RP}}^{\frac{1}{3K+2}},$$

where $\epsilon_{\mathsf{RP}} := \frac{\log(NH|\mathcal{R}||\mathcal{P}|/\delta)}{M}$ and then we have for all $\mu_i \in \Pi_i(C)$ that

$$\mathbb{E}_{c \sim \rho} \left[ V_{i,1}^{\mu_i \circ \widehat{\pi}_{-i}, r^\star}(c, \boldsymbol{s}_1) \right] - \mathbb{E}_{c \sim \rho} \left[ V_{i,1}^{\widehat{\pi}, r^\star}(c, \boldsymbol{s}_1) \right] \lesssim CC_{\mathsf{S}}^{\frac{K}{3K+2}} H^{\frac{6K+2}{3K+2}} (2N^2)^{\frac{K-1}{3K+2}} \epsilon_{\mathsf{RP}}^{\frac{1}{3K+2}}.$$

This concludes our proof.

### F.1 PROOF OF LEMMA 9

Note that given $c$, the distribution of $(s_j, a_j)$ is independent from each other due to the decoupled transition. Therefore we have

$$\mathbb{E}_{c \sim \rho} \left[ \sum_{\boldsymbol{s}, \boldsymbol{a}} \left| d_h^\pi(\boldsymbol{s}, \boldsymbol{a}|c) - \widehat{d}_h^\pi(\boldsymbol{s}, \boldsymbol{a}|c) \right| \right] = \mathbb{E}_{c \sim \rho} \left[ \sum_{\boldsymbol{s}, \boldsymbol{a}} \left| \prod_{j \in [N]} d_h^{\pi_j}(s_j, a_j|c) - \prod_{j \in [N]} \widehat{d}_h^{\pi_j}(s_j, a_j|c) \right| \right]$$

Now for any $0 \leq k \leq N - 1$, consider the following difference:

$$I_k := \mathbb{E}_{c \sim \rho} \left[ \sum_{\boldsymbol{s}, \boldsymbol{a}} \left| \prod_{1 \leq j \leq k} d_h^{\pi_j}(s_j, a_j|c) \prod_{k+1 \leq j \leq N} \widehat{d}_h^{\pi_j}(s_j, a_j|c) - \prod_{1 \leq j \leq k+1} d_h^{\pi_j}(s_j, a_j|c) \prod_{k+2 \leq j \leq N} \widehat{d}_h^{\pi_j}(s_j, a_j|c) \right| \right]$$

Note that we have

$$I_k = \mathbb{E}_{c \sim \rho} \left[ \sum_{\boldsymbol{s}, \boldsymbol{a}} \prod_{1 \leq j \leq k} d_h^{\pi_j}(s_j, a_j|c) \prod_{k+2 \leq j \leq N} \widehat{d}_h^{\pi_j}(s_j, a_j|c) \left| \widehat{d}_h^{\pi_{k+1}}(s_{k+1}, a_{k+1}|c) - d_h^{\pi_{k+1}}(s_{k+1}, a_{k+1}|c) \right| \right]$$

$$= \mathbb{E}_{c \sim \rho} \left[ \sum_{s_{k+1}, a_{k+1}} \left| \widehat{d}_h^{\pi_{k+1}}(s_{k+1}, a_{k+1}|c) - d_h^{\pi_{k+1}}(s_{k+1}, a_{k+1}|c) \right| \right.$$

$$
\cdot \sum_{\boldsymbol{s}_{-(k+1)}, \boldsymbol{a}_{-(k+1)}} \prod_{1 \leq j \leq k} d_h^{\pi_j}(s_j, a_j | c) \prod_{k+2 \leq j \leq N} \widehat{d}_h^{\pi_j}(s_j, a_j | c) \Bigg]
$$

$$
= \mathbb{E}_{c \sim \rho} \left[ \sum_{s_{k+1}, a_{k+1}} \left| \widehat{d}_h^{\pi_{k+1}}(s_{k+1}, a_{k+1} | c) - d_h^{\pi_{k+1}}(s_{k+1}, a_{k+1} | c) \right| \right]
$$

Therefore we have

$$
\mathbb{E}_{c \sim \rho} \left[ \sum_{\boldsymbol{s}, \boldsymbol{a}} \left| d_h^{\pi}(\boldsymbol{s}, \boldsymbol{a} | c) - \widehat{d}_h^{\pi}(\boldsymbol{s}, \boldsymbol{a} | c) \right| \right] \leq \sum_{k=0}^{N-1} I_k = \sum_{j=1}^{N} \mathbb{E}_{c \sim \rho} \left[ \sum_{s_j, a_j} \left| d_h^{\pi_j}(s_j, a_j | c) - \widehat{d}_h^{\pi_j}(s_j, a_j | c) \right| \right].
$$

This concludes our proof.

### F.2 PROOF OF LEMMA 10

Let $\delta_h$ denote $\mathbb{E}_{c \sim \rho} \left[ \sum_{s_j, a_j} \left| d_h^{\pi_j}(s_j, a_j | c) - \widehat{d}_h^{\pi_j}(s_j, a_j | c) \right| \right]$. Then we know $\delta_1 = 0$. In addition, for any $1 \leq h \leq H$, we have

$$
\delta_h = \mathbb{E}_{c \sim \rho} \Bigg[ \sum_{s_j, a_j} \Bigg| \sum_{s_j', a_j'} d_{h-1}^{\pi_j}(s_j', a_j' | c) P_{j,h}(s_j | c, s_j', a_j') \pi_{j,h}(a_j | c, s_j)
$$

$$
- \sum_{s_j', a_j'} \widehat{d}_{h-1}^{\pi_j}(s_j', a_j' | c) \widehat{P}_{j,h}(s_j | c, s_j', a_j') \pi_{j,h}(a_j | c, s_j) \Bigg| \Bigg]
$$

$$
= \mathbb{E}_{c \sim \rho} \Bigg[ \sum_{s_j, a_j} \Bigg| \Bigg( \sum_{s_j', a_j'} d_{h-1}^{\pi_j}(s_j', a_j' | c) P_{j,h}(s_j | c, s_j', a_j') \pi_{j,h}(a_j | c, s_j)
$$

$$
- \sum_{s_j', a_j'} d_{h-1}^{\pi_j}(s_j', a_j' | c) \widehat{P}_{j,h}(s_j | c, s_j', a_j') \pi_{j,h}(a_j | c, s_j) \Bigg)
$$

$$
+ \Bigg( \sum_{s_j', a_j'} d_{h-1}^{\pi_j}(s_j', a_j' | c) \widehat{P}_{j,h}(s_j | c, s_j', a_j') \pi_{j,h}(a_j | c, s_j)
$$

$$
- \sum_{s_j', a_j'} \widehat{d}_{h-1}^{\pi_j}(s_j', a_j' | c) \widehat{P}_{j,h}(s_j | c, s_j', a_j') \pi_{j,h}(a_j | c, s_j) \Bigg) \Bigg| \Bigg]
$$

$$
\leq \mathbb{E}_{c \sim \rho} \Bigg[ \sum_{s_j, a_j} \Bigg| \sum_{s_j', a_j'} \Big( d_{h-1}^{\pi_j}(s_j', a_j' | c) P_{j,h}(s_j | c, s_j', a_j') \pi_{j,h}(a_j | c, s_j)
$$

$$
- d_{h-1}^{\pi_j}(s_j', a_j' | c) \widehat{P}_{j,h}(s_j | c, s_j', a_j') \pi_{j,h}(a_j | c, s_j) \Big) \Bigg| \Bigg]
$$

$$
+ \mathbb{E}_{c \sim \rho} \Bigg[ \sum_{s_j, a_j} \Bigg| \sum_{s_j', a_j'} \Big( d_{h-1}^{\pi_j}(s_j', a_j' | c) \widehat{P}_{j,h}(s_j | c, s_j', a_j') \pi_{j,h}(a_j | c, s_j)
$$

$$
- \widehat{d}_{h-1}^{\pi_j}(s_j', a_j' | c) \widehat{P}_{j,h}(s_j | c, s_j', a_j') \pi_{j,h}(a_j | c, s_j) \Big) \Bigg| \Bigg]
$$

$$
\leq \mathbb{E}_{c \sim \rho} \Bigg[ \sum_{s_j, a_j, s_j', a_j'} d_{h-1}^{\pi_j}(s_j', a_j' | c) \pi_{j,h}(a_j | c, s_j) \Big| P_{j,h}(s_j | c, s_j', a_j') - \widehat{P}_{j,h}(s_j | c, s_j', a_j') \Big| \Bigg]
$$

$$
+ \mathbb{E}_{c \sim \rho} \Bigg[ \sum_{s_j, a_j, s_j', a_j'} \widehat{P}_{j,h}(s_j | c, s_j', a_j') \pi_{j,h}(a_j | c, s_j) \Big| d_{h-1}^{\pi_j}(s_j', a_j' | c) - \widehat{d}_{h-1}^{\pi_j}(s_j', a_j' | c) \Big| \Bigg]
$$

$$
= \mathbb{E}_{c\sim\rho}\left[\sum_{s_j,a_j,s_j'} d_{h-1}^{\pi_j}(s_j',a_j'|c)\left|P_{j,h}(s_j|c,s_j',a_j') - \widehat{P}_{j,h}(s_j|c,s_j',a_j')\right|\sum_{a_j}\pi_{j,h}(a_j|c,s_j)\right]
$$

$$
+ \mathbb{E}_{c\sim\rho}\left[\sum_{s_j',a_j'}\left|d_{h-1}^{\pi_j}(s_j',a_j'|c) - \widehat{d}_{h-1}^{\pi_j}(s_j',a_j'|c)\right|\sum_{s_j,a_j}\widehat{P}_{j,h}(s_j|c,s_j',a_j')\pi_{j,h}(a_j|c,s_j)\right]
$$

$$
= \mathbb{E}_{c\sim\rho}\left[\sum_{s_j,a_j,s_j'} d_{h-1}^{\pi_j}(s_j',a_j'|c)\left|P_{j,h}(s_j|c,s_j',a_j') - \widehat{P}_{j,h}(s_j|c,s_j',a_j')\right|\sum_{a_j}\pi_{j,h}(a_j|c,s_j)\right]
$$

$$
+ \mathbb{E}_{c\sim\rho}\left[\sum_{s_j',a_j'}\left|d_{h-1}^{\pi_j}(s_j',a_j'|c) - \widehat{d}_{h-1}^{\pi_j}(s_j',a_j'|c)\right|\sum_{s_j,a_j}\widehat{P}_{j,h}(s_j|c,s_j',a_j')\pi_{j,h}(a_j|c,s_j)\right]
$$

$$
= \mathbb{E}_{c\sim\rho,(s_j,a_j)\sim d_{h-1}^{\pi_j}(\cdot|c)}\left[\left\|\widehat{P}_{j,h-1}(\cdot|c,s_j,a_j) - P_{j,h-1}^\star(\cdot|c,s_j,a_j)\right\|_1\right] + \delta_{h-1}.
$$

Therefore, we have

$$
\mathbb{E}_{c\sim\rho}\left[\sum_{s_j,a_j}\left|d_h^{\pi_j}(s_j,a_j|c) - \widehat{d}_h^{\pi_j}(s_j,a_j|c)\right|\right]
$$

$$
\leq \sum_{h'=1}^{h-1}\mathbb{E}_{c\sim\rho,(s_j,a_j)\sim d_{h'}^{\pi_j}(\cdot|c)}\left[\left\|\widehat{P}_{j,h'}(\cdot|c,s_j,a_j) - P_{j,h'}^\star(\cdot|c,s_j,a_j)\right\|_1\right].
$$

This concludes our proof.

### F.3 PROOF OF LEMMA 11

First it can be observed that $\widehat{V}_{i,h}^{\mu_i\circ\pi_{-i},\widehat{r}}(c,s_{i,h}) = \mathbb{E}_{\boldsymbol{s}_{-i}\sim\widehat{d}_h^{\pi_{-i}}}\left[\widehat{V}_{i,h}^{\mu_i\circ\pi_{-i},\widehat{r}}(c,\boldsymbol{s}_h)\right]$. Note that we have

$$
\widehat{Q}_{i,h}^{\mu_i\circ\pi_{-i},\widehat{r}}(c,s_{i,h},a_{i,h}) = \mathbb{E}_{(\boldsymbol{s}_{-i,h},\boldsymbol{a}_{-i,h})\sim\widehat{d}_h^{\pi_{-i}}(\cdot|c)}\left[\mathbb{E}_{\mu_i\circ\pi_{-i},\widehat{P}}\left[\sum_{h'=h}^H \widehat{r}_{i,h'}(c,\boldsymbol{s}_{h'},\boldsymbol{a}_{h'})\Big|c,\boldsymbol{s}_h,\boldsymbol{a}_h\right]\right]
$$

$$
= \mathbb{E}_{(\boldsymbol{s}_{-i,h},\boldsymbol{a}_{-i,h})\sim\widehat{d}_h^{\pi_{-i}}(\cdot|c)}\left[\widehat{r}_{i,h}(c,\boldsymbol{s}_h,\boldsymbol{a}_h)\right]
$$

$$
+ \mathbb{E}_{(\boldsymbol{s}_{-i,h},\boldsymbol{a}_{-i,h})\sim\widehat{d}_h^{\pi_{-i}}(\cdot|c)}\left[\mathbb{E}_{\mu_i\circ\pi_{-i},\widehat{P}}\left[\sum_{h'=h+1}^H \widehat{r}_{i,h}(c,\boldsymbol{s}_{h'},\boldsymbol{a}_{h'})\Big|c,\boldsymbol{s}_h,\boldsymbol{a}_h\right]\right],
$$

where we use $\mathbb{E}_{\pi,\widehat{P}}[\cdot]$ to denote the distribution of the trajectory when executing joint policy $\pi$ with transition model $\widehat{P}$.

On the other hand we know

$$
\mathbb{E}_{\mu_i\circ\pi_{-i},\widehat{P}}\left[\sum_{h'=h+1}^H \widehat{r}_{i,h}(c,\boldsymbol{s}_{h'},\boldsymbol{a}_{h'})\Big|c,\boldsymbol{s}_h,\boldsymbol{a}_h\right]
$$

$$
= \mathbb{E}_{s_{j,h+1}\sim\widehat{P}_{j,h}(\cdot|c,s_{j,h},a_{j,h}),\forall j}\left[\mathbb{E}_{\mu_i\circ\pi_{-i},\widehat{P}}\left[\sum_{h'=h+1}^H \widehat{r}_{i,h}(c,\boldsymbol{s}_{h'},\boldsymbol{a}_{h'})\Big|c,\boldsymbol{s}_{h+1}\right]\right]
$$

$$
= \mathbb{E}_{s_{j,h+1}\sim\widehat{P}_{j,h}(\cdot|c,s_{j,h},a_{j,h}),\forall j}\left[\widehat{V}_{i,h+1}^{\mu_i\circ\pi_{-i},\widehat{r}}(c,\boldsymbol{s}_{h+1})\right].
$$

Therefore we know

$$
\widehat{Q}_{i,h}^{\mu_i\circ\pi_{-i},\widehat{r}}(c,s_{i,h},a_{i,h})
$$

$$
= \mathbb{E}_{(\boldsymbol{s}_{-i,h},\boldsymbol{a}_{-i,h})\sim\widehat{d}_h^{\pi_{-i}}(\cdot|c)}\left[\widehat{r}_{i,h}(c,\boldsymbol{s}_h,\boldsymbol{a}_h)\right]
$$

$$
+ \mathbb{E}_{s_{i,h+1}\sim\widehat{P}_{i,h}(\cdot|c,s_{i,h},a_{i,h}),(\boldsymbol{s}_{-i,h},\boldsymbol{a}_{-i,h})\sim\widehat{d}_h^{\pi_{-i}}(\cdot|c),s_{j,h+1}\sim\widehat{P}_{j,h}(\cdot|c,s_{j,h},a_{j,h}),\forall j\neq i}\left[\widehat{V}_{i,h+1}^{\mu_i\circ\pi_{-i},\widehat{r}}(c,\boldsymbol{s}_{h+1})\right]
$$

$$= \mathbb{E}_{(\boldsymbol{s}_{-i,h},\boldsymbol{a}_{-i,h})\sim \widehat{d}_h^{\pi-i}(\cdot|c)} \left[ \widehat{r}_{i,h}(c,\boldsymbol{s}_h,\boldsymbol{a}_h) \right] + \mathbb{E}_{s_{i,h+1}\sim \widehat{P}_{i,h}(\cdot|c,s_{i,h},a_{i,h}),\boldsymbol{s}_{-i,h+1}\sim \widehat{d}_{h+1}^{\pi-i}(\cdot|c)} \left[ \widehat{V}_{i,h+1}^{\mu_i \circ \pi_{-i},\widehat{r}}(c,\boldsymbol{s}_{h+1}) \right]$$

$$= \mathbb{E}_{(\boldsymbol{s}_{-i,h},\boldsymbol{a}_{-i,h})\sim \widehat{d}_h^{\pi-i}(\cdot|c)} \left[ \widehat{r}_{i,h}(c,\boldsymbol{s}_h,\boldsymbol{a}_h) \right] + \mathbb{E}_{s_{i,h+1}\sim \widehat{P}_{i,h}(\cdot|c,s_{i,h},a_{i,h})} \left[ \widehat{V}_{i,h}^{\mu_i \circ \pi_{-i},\widehat{r}}(c,s_{i,h+1}) \right].$$

This concludes our proof.

### F.4 PROOF OF LEMMA 12

Let $\widetilde{r}_{i,h}(c,s_i,a_i)$ denote $\mathbb{E}_{(\boldsymbol{s}_{-i},\boldsymbol{a}_{-i})\sim d_h^{\pi-i}(\cdot|c)} [r_{i,h}(c,\boldsymbol{s},\boldsymbol{a})]$. From Lemma 11, we have

$$V_{i,1}^{\mu_i' \circ \pi_{-i},r}(c,s_{i,1}) - V_{i,1}^{\mu_i \circ \pi_{-i},r}(c,s_{i,1}) = \mathbb{E}_{\mu_i'} \left[ \sum_{h=1}^{H} \widetilde{r}_{i,h}(c,s_{i,h},a_{i,h}) \Big| c \right] - V_{i,1}^{\mu_i \circ \pi_{-i},r}(c,s_{i,1})$$

$$= \mathbb{E}_{\mu_i'} \left[ \sum_{h=2}^{H} \widetilde{r}_{i,h}(c,s_{i,h},a_{i,h}) \Big| c \right] + \mathbb{E}_{\mu_i'} \left[ \widetilde{r}_{i,1}(s_{i,1},a_{i,1}) - V_{i,1}^{\mu_i \circ \pi_{-i},r}(c,s_{i,1}) \Big| c \right]$$

$$= \mathbb{E}_{\mu_i'} \left[ \sum_{h=2}^{H} \widetilde{r}_{i,h}(c,s_{i,h},a_{i,h}) \Big| c \right] + \mathbb{E}_{\mu_i'} \left[ Q_{i,1}^{\mu_i \circ \pi_{-i},r}(c,s_{i,1},a_{i,1}) - V_{i,2}^{\mu_i \circ \pi_{-i},r}(c,s_{i,2}) - V_{i,1}^{\mu_i \circ \pi_{-i},r}(c,s_{i,1}) \Big| c \right]$$

$$= \mathbb{E}_{\mu_i'} \left[ \sum_{h=2}^{H} \widetilde{r}_{i,h}(c,s_{i,h},a_{i,h}) \Big| c \right] - \mathbb{E}_{\mu_i'} \left[ V_{i,2}^{\mu_i \circ \pi_{-i},r}(c,s_{i,2}) \right]$$

$$+ \mathbb{E}_{s_{i,1}\sim d_1^{\mu_i'}(\cdot|c)} \left[ \left\langle Q_{i,1}^{\mu_i \circ \pi_{-i},r}(c,s_{i,1},\cdot), \mu_{i,1}'(\cdot|c,s_{i,1}) - \mu_{i,1}(\cdot|c,s_{i,1}) \right\rangle \right].$$

Here the first step is due to the definition of value function and the third step is due to Lemma 11. Now apply the above arguments recursively to $\mathbb{E}_{\mu_i'} \left[ \sum_{h=2}^{H} \widetilde{r}_{i,h}(c,s_{i,h},a_{i,h}) \Big| c \right] - \mathbb{E}_{\mu_i'} \left[ V_{i,2}^{\mu_i \circ \pi_{-i},r}(c,s_{i,2}) \right]$ and we have

$$V_{i,1}^{\mu_i' \circ \pi_{-i},r}(c,s_{i,1}) - V_{i,1}^{\mu_i \circ \pi_{-i},r}(c,s_{i,1}) = \sum_{h=1}^{H} \mathbb{E}_{s_{i,h}\sim d_h^{\mu_i'}(\cdot|c)} \left[ \left\langle Q_{i,h}^{\mu_i \circ \pi_{-i},r}(c,s_{i,h},\cdot), \mu_{i,h}'(\cdot|c,s_{i,h}) - \mu_{i,h}(\cdot|c,s_{i,h}) \right\rangle \right].$$

This concludes our proof.

# G    AUXILIARY LEMMAS

**Lemma 13** (Song et al. (2022)). *Let $\{(x_m, y_m)\}_{m=1}^M$ be $M$ samples that are independently sampled from $x_m \sim p$ and $y_m \sim q(\cdot|x_m) := f^\star(x_m) + \epsilon_m$ where $\epsilon_m$ is a random noise. Suppose that $y_m \in [0, 1]$ for all $m \in [M]$ and we have access to a function class $\mathcal{G} : \mathcal{X} \to [0, 1]$ which satisfies $f^\star \in \mathcal{G}$. Then if $\{\epsilon_m\}_{m=1}^M$ are independent and $\mathbb{E}[y_m|x_m] = f^\star(x_m)$, we have with probability at least $1 - \delta$ that*

$$\mathbb{E}_{x \sim p}[(\widehat{f}(x) - f^\star(x))^2] \lesssim \frac{\log(|\mathcal{G}|/\delta)}{M},$$

*where $\widehat{f} = \arg\min_{f \in \mathcal{G}} \sum_{m=1}^M (f(x_m) - y_m)^2$ is the LSR solution.*

**Lemma 14** (Zhan et al. (2023b)). *Let $\{(x_m, y_m)\}_{m=1}^M$ be $M$ samples that are i.i.d. sampled from $x_m \sim p$ and $y_m \sim q^\star(\cdot|x_m)$. Suppose we have access to a probability model class $\mathcal{Q}$ which satisfies $q^\star \in \mathcal{Q}$. Then we have with probability at least $1 - \delta$ that*

$$\mathbb{E}_{x \sim p}\left[\|\widehat{q}(\cdot|x) - q^\star(\cdot|x)\|_1^2\right] \lesssim \frac{\log(|\mathcal{Q}|/\delta)}{M},$$

*where $\widehat{q} = \arg\min_{q \in \mathcal{Q}} \sum_{m=1}^M \log q(y_m|x_m)$ is the MLE solution.*

