# OpenReview forum: "Exploiting Structure in Offline Multi-Agent RL: The Benefits of Low Interaction Rank"
_ICLR.cc/2025/Conference — ICLR 2025 Poster_

### Official Review · Reviewer_oM8M · 2024-10-22

**Soundness:** 3
**Presentation:** 3
**Contribution:** 3
**Rating:** 6
**Confidence:** 3

**Summary:**

The paper studies learning a coarse correlated equilibrium in the offline multi-agent reinforcement learning. By imposing assumptions on the interaction rank of reward models, the paper establishes a sample complexity bound that grows exponentially with respect to the interaction rank, instead of the number of agents. Finally, the author use numerical experiments to examine the effectiveness of the proposed algorithm.

**Strengths:**

* The paper is the first that proposes a decentralized algorithm for offline general-sum MARL games and establishes its probable statistical guarantees.
* The integration of interaction rank to MARL provides an interesting yet also technically non-trivial angle.
* The paper flow is good, and the author uses contextual games to serve as a toy example to help readers understand the idea of the paper. I like the proof sketch provided for the contextual game when K = 2.

**Weaknesses:**

* The authors are encouraged to release the codes of the numerical experiments.
* Just a suggestion, maybe the author can consider using Markov Game or MG to replace RL in the title or in the abstract. I feel like people usually refers to the cooperative setting when saying MARL.
* The considered problems are contextual game, which does not have the notion of state, and transition-decoupled MG. I believe the author could directly brought up the transition decouple assumption in the problem formulation, since it is an assumption on the setting instead of technical assumptions.

**Questions:**

* I wonder does similar property applies to the cooperative setting? I would like to hear the author's insight on the difference between these two settings (cooperative and competitive).
* In MARL, another widely-used property is the exponential decay of transition dynamics, e.g., [1,2]. From a intuitive perspective, is "low interaction rank" basically assumes a similar property, instead of on the transition dynamics, but on the reward structure? If they do share relevant ideas, I suggest the authors also discuss in the paper. Also, do you think the exponential decay property can be used to alleviate the current decoupled transition property in the paper?

[1] Qu, G., Wierman, A., & Li, N. (2022). Scalable reinforcement learning for multiagent networked systems. Operations Research, 70(6), 3601-3628.

[2] Ying, D., Zhang, Y., Ding, Y., Koppel, A., & Lavaei, J. (2024). Scalable primal-dual actor-critic method for safe multi-agent rl with general utilities. Advances in Neural Information Processing Systems, 36.

---

> ### Author Response · Authors · 2024-11-15
> **Response**
>
> Thank you for your review! Here are our response:
>
> 1. **Release the code**
>
> We will attach a link to the codes in another comment soon. Upon acceptance, we will clean it up for a release.
>
>
> 2. **Replace MARL**
>
> Thank you for the suggestion! There are indeed some papers in MARL that consider the competitive setting [1-3]. We will clarify this in the abstract, namely, mention our work is relevant for both the cooperative and competitive settings.
>
>
>
> 3. **Move decoupled transition to the preliminaries**
>
> Thank you for the suggestion! We will move it in the revised version.
>
>
>
> 4. **Difference between cooperative setting and competitive setting**
>
> Our discussion applies to both cooperative and competitive settings since we are considering general-sum MGs. As a matter of fact, the 1-IR critics have been known as VDNs and widely utilized in empirical online cooperative MARL literature. That said, one key difference is that 1-IR critics perform poorly in the offline setting, as shown in Section 6. This is because in the online cooperative setting, the agent can continually collect fresh samples to update the estimated 1-IR reward so that the critic can learn accurate local approximations of the current expected reward even if the other agents’ policies change. However, in the offline setting, a 1-IR critic cannot make such updates because iterative data collection is not allowed. This problem is further worsened in competitive games because each agent has different objectives. Therefore, in the offline competitive setting, 1-IR critic models can be severely biased and we should use critics with a higher IR to capture the interaction among the agents.
>
>
>
> 5. **Exponential decay**
>
> We would like to clarify that IR is different from the exponential decay in the existing works. Exponential decay means that each agent’s Q function only depends on the state and action of its $k$-hop neighbors and itself approximately. However, in IR, each agent’s reward function can depend on the state and action of all agents. We only require that the joint interaction can be decomposed to a group of partial interactions. These partial interactions can involve all agents in general.
>
> That said, we think that the exponential decay might be able to help alleviate the decoupled transition assumption, but each agent would need to learn a critic that depends on the states and actions of all $k$-hop neighbors. This is difficult in the decentralized setting because the agents do not have access to other agents’ states and actions.
>
> [1] Lowe, R., Wu, Y. I., Tamar, A., Harb, J., Pieter Abbeel, O., & Mordatch, I. (2017). Multi-agent actor-critic for mixed cooperative-competitive environments. Advances in neural information processing systems, 30.
>
> [2] Li, W., Ding, Z., Karten, S., & Jin, C. (2024). FightLadder: A Benchmark for Competitive Multi-Agent Reinforcement Learning. arXiv preprint arXiv:2406.02081.
>
> [3] Competitive Multi-Agent Reinforcement Learning (DDPG) with TorchRL Tutorial — torchrl main documentation (pytorch.org)

---

> ### Author Response · Authors · 2024-11-24
> **Additional Questions?**
>
> Dear reviewer,
>
> We have uploaded a revised version of our paper where we clarified the cooperative/competitive MARL and moved decoupled transition to the preliminaries. Since the discussion period is ending soon, we would like to kindly inquire whether our response has addressed your concerns. If you have additional questions, please do not hesitate to let us know. Thank you once again for your valuable feedback.

---

> > ### Comment · Reviewer_oM8M · 2024-11-28
> >
> > Thank you for your responses. I am happy to maintain my scores.

---

### Official Review · Reviewer_MR2u · 2024-11-01

**Soundness:** 3
**Presentation:** 3
**Contribution:** 3
**Rating:** 6
**Confidence:** 2

**Summary:**

The paper first shows that if the underlying problem has low interaction rank then it is robust to distribution shifts coming from the mismatch between training and target distributions. Then the authors propose a decentralised algorithm for multi-agent offline reinforcement learning and explicitly derive a performance guarantee that depends on the interaction rank of the function K. The theoretical results are verified with experiments on a toy problem.

**Strengths:**

- The paper states a clear research question and draws clear conclusions that explicitly show low interaction rank games are easier to learn. While this result is not surprising, it is good that the paper explicitly derives it and quantifies this dependence.
-  The paper proposes a novel algorithm, which is decentralised and can learn from offline data.

**Weaknesses:**

- The empirical evaluation is limited to one toy problem only, with no real-world experiments
- The paper could use a more explicit related work section. For example, currently when reading the paper it is not clear what is the state of the art when it comes to offline, decentralised, MARL and how does the proposed algorithm compares to it.

**Questions:**

- The authors considered specifically the decentralised setting. I wonder if a better performance guarantee can be obtained if we assumed centralised setting.

---

> ### Author Response · Authors · 2024-11-15
> **Response**
>
> Thank you for your review! Here are our response:
>
> 1. **Real-world experiments**
>
> In this work, we hope to help lay out theoretical foundations for more powerful future offline MARL algorithms. As such, we focus on our key theoretical results and use the empirical evaluation to validate that these ideas (i.e., low-IR critics) extend to practical implementations of offline MARL algorithms. While we acknowledge that real-world experiments would further strengthen our results, we also believe that this is beyond the scope of this current study.
>
> 2. **Additional related works**
>
> We would like to point out that for offline general-sum Markov games, there are no existing decentralized and provably efficient algorithms and this field is under-explored. Our work can be viewed as a first attempt towards this objective. For more background information, the related works are stated in the following.
>
> - **Provable Offline  Markov Games**
>
> [1] studies offline tabular zero-sum Markov games and [2]  improves the complexity to minimax optimal. [3-4] further extends the discussion to linear function approximation. However, the above works are all limited to two-agent zero-sum games. For multi-agent general-sum Markov games, the only existing works are [5-6], where [5] investigates the tabular setting and later [6] proposes a centralized algorithm with function approximation. Nevertheless, both of them are centralized and computationally inefficient since they need to evaluate each possible joint policy, whose complexity will scale with the number of agents exponentially. In comparison, our method is decentralized and computationally feasible.
>
> - **Decentralized Algorithms in general-sum MGs**
>
> In the online setting, to learn a CCE for a general-sum MG, [10-12] proposed V-learning and [13] introduced SPoMAR. However, these algorithms cannot achieve no-regret and depend heavily on collecting fresh samples, thus cannot be extended to the offline setting trivially.
>
> For specific no-regret decentralized algorithms, as far as we know, [14] is the only existing decentralized work which achieves sublinear regret in general-sum MGs (and thus is able to learn a CCE). Nevertheless, they only focus on tabular cases in the full information setting or online setting with a minimum reachability assumption, which are drastically different from the offline setting.
>
> - **Factored Critics**
>
> Another closely related line of work is the utilization of factored critics in empirical studies. Since [7] proposed the value decomposition networks (VDN), factored critics have been widely applied in the online cooperative MARL setting. [8] further proposed QMIX, a Q-learning type algorithm which utilizes a non-linear mixing network to combine single-agent critics, and [9] extended the discussion into actor-critic methods. Nevertheless, most of these works focus on online cooperative MARL and lack theoretical analysis. Our paper indeed formalizes and generalizes these efforts by introducing the interaction-rank structure. As a matter of fact, VDNs are exactly a 1-IR critic. We further show that the idea of factorization in critics is also useful in offline general-sum MGs, but in general you need critics with a higher IR than VDNs for offline learning, as discussed in Section 6.
>
> 3. **Centralized setting**
>
> We think centralized training might improve the sample complexity. However, the computation cost will increase since in the centralized setting one usually trains a joint value function. In addition, decentralized algorithms are more desirable when the number of agents is large due to their scalability.

---

> > ### Author Response · Authors · 2024-11-15
> > **References**
> >
> > [1] Qiwen Cui and Simon S Du. When is offline two-player zero-sum markov game solvable? arXiv preprint arXiv:2201.03522, 2022a.
> >
> > [2] Yuling Yan, Gen Li, Yuxin Chen, and Jianqing Fan. Model-based reinforcement learning is minimax-optimal for offline zero-sum markov games. arXiv preprint arXiv:2206.04044, 2022.
> >
> > [3] Han Zhong,Wei Xiong, Jiyuan Tan, LiweiWang, Tong Zhang, ZhaoranWang, and Zhuoran Yang. Pessimistic minimax value iteration: Provably efficient equilibrium learning from offline datasets. arXiv preprint arXiv:2202.07511, 2022.
> >
> > [4] Wei Xiong, Han Zhong, Chengshuai Shi, Cong Shen, Liwei Wang, and Tong Zhang. Nearly minimax optimal offline reinforcement learning with linear function approximation: Single-agent mdp and markov game. arXiv preprint arXiv:2205.15512, 2022.
> >
> > [5] Qiwen Cui and Simon S Du. Provably efficient offline multi-agent reinforcement learning via
> > strategy-wise bonus. Advances in Neural Information Processing Systems, 35:11739–11751, 2022.
> >
> > [6] Yuheng Zhang, Yu Bai, and Nan Jiang. Offline learning in markov games with general function approximation. In International Conference on Machine Learning, pp. 40804–40829. PMLR, 2023b.
> >
> > [7] Peter Sunehag, Guy Lever, Audrunas Gruslys, Wojciech Marian Czarnecki, Vinícius Flores Zambaldi, Max Jaderberg, Marc Lanctot, Nicolas Sonnerat, Joel Z Leibo, Karl Tuyls, et al. Value-decomposition networks for cooperative multi-agent learning based on team reward. In AAMAS, pages 2085–2087, 2018.
> >
> > [8] Tabish Rashid, Mikayel Samvelyan, Christian SchroederWitt, Gregory Farquhar, Jakob Foerster, and Shimon Whiteson. Qmix: Monotonic value function factorisation for deep multi-agent reinforcement learning. In International Conference on Machine Learning, pages 4292–4301, 2018.
> >
> > [9] Peng, B., Rashid, T., Schroeder de Witt, C., Kamienny, P. A., Torr, P., Böhmer, W., & Whiteson, S. (2021). Facmac: Factored multi-agent centralised policy gradients. Advances in Neural Information Processing Systems, 34, 12208-12221.
> >
> > [10] Jin, C., Liu, Q., Wang, Y., and Yu, T. V-learning–a simple, efficient, decentralized algorithm for multiagent RL. arXiv preprint arXiv:2110.14555, 2021.
> >
> >
> > [11] Song, Z., Mei, S., and Bai, Y. When can we learn general-sum markov games with a large number of players sample-efficiently? In International Conference on Learning Representations, 2022.
> >
> >
> > [12] Mao, W. and Basar, T. Provably efficient reinforcement learning in decentralized general-sum markov games. CoRR, abs/2110.05682, 2021.
> >
> >
> > [13] Daskalakis, C., Golowich, N., and Zhang, K. The complexity of markov equilibrium in stochastic games, 2022.
> >
> > [14] Erez, L., Lancewicki, T., Sherman, U., Koren, T., and Mansour, Y. Regret minimization and convergence to equilibria in general-sum markov games, 2022.

---

> > ### Comment · Reviewer_MR2u · 2024-11-21
> >
> > Thank you for your response. I believe point 2 of your response should be added to the paper. It really helps to clarify what is the relationship of your work to the existing literature. Overall, I remain positive about the paper.

---

> > > ### Author Response · Authors · 2024-11-24
> > >
> > > We have added more related works to the paper in the revised version. Thank you once again for your valuable feedback!

---

### Official Review · Reviewer_8HY6 · 2024-11-03

**Soundness:** 3
**Presentation:** 3
**Contribution:** 3
**Rating:** 8
**Confidence:** 3

**Summary:**

This paper introduces a structural assumption -- the interaction rank (IR), and establishes that functions with low interaction rank are more robust to distribution shift compared to general ones. The authors show that learning an approximate equilibrium in offline MARL can scale exponentially with the IR instead of exponentially with the number of agents. The proposed algorithm is a decentralized, no-regret learning algorithm that can be implemented in practical settings while utilizing standard RL algorithms.

**Strengths:**

* The paper leverages a structural assumption that shows the potential of using reward architectures with low interaction rank in offline MARL setting with an orderly better sample complexity, which seems to be a promising future direction.
* The paper is well-written and easy to follow.

**Weaknesses:**

* I don't see much discussion on related MARL or RL literature regarding interaction rank or similar assumptions. Discussion on literature of related topic would help determine the novelty of the idea and quantify the contribution.

**Questions:**

* By leveraging the low interaction rank assumption, the proposed algorithm achieves a sample complexity that scales exponentially
 with the interaction rank instead of number of agents. Meanwhile, there exists a constant as the coefficient of the sample complexity that hides behind the operator "$\lesssim$". How to understand the constant? Does it scales with the problem size?
* In extreme case where the problem has full interaction rank, does the proposed sample complexity orderly equal to the case that scales with number of agents? If not what makes the difference?

---

> ### Author Response · Authors · 2024-11-15
> **Response**
>
> Thank you for your review! Here are our response:
> 1. **Related works**
>
> Thank you for pointing this out! We will include the following related works in the revised version.
>
>  - **Provable Offline  Markov Games**
>
> [1] studies offline tabular zero-sum Markov games and [2]  improves the complexity to minimax optimal. [3-4] further extends the discussion to linear function approximation. However, the above works are all limited to two-agent zero-sum games. For multi-agent general-sum Markov games, the only existing works are [5-6], where [5] investigates the tabular setting and later [6] proposes a centralized algorithm with function approximation. Nevertheless, both of them are centralized and computationally inefficient since they need to evaluate each possible joint policy, whose complexity will scale with the number of agents exponentially. In comparison, our method is decentralized and computationally feasible.
>
> - **Factored Critics**
>
> Another closely related line of work is the utilization of factored critics in empirical studies. Since [7] proposed the value decomposition networks (VDN), factored critics have been applied in the online cooperative MARL setting. [8] further proposed QMIX, a Q-learning type algorithm which utilizes a non-linear mixing network to combine single-agent critics, and [9] extended the discussion into actor-critic methods. Nevertheless, most of these works focus on online cooperative MARL and lack theoretical analysis. Our paper indeed formalizes and generalizes these efforts by introducing the interaction-rank structure. As a matter of fact, VDNs are exactly a 1-IR critic. We further show that the idea of factorization in critics is also useful in offline general-sum MGs, but in general you need critics with a higher IR than VDNs for offline learning, as discussed in Section 6.
>
> 2. **Hiding constant**
>
> This constant is universal and independent from the problem size such as the number of agents $N$ and the size of the function classes.
>
> 3. **Where the problem has full interaction rank**
>
> Yes, it will scale exponentially with the number of agents. This is also the reason why we emphasize the importance of utilizing a low-IR structure for the critics.
>
> [1] Qiwen Cui and Simon S Du. When is offline two-player zero-sum markov game solvable? arXiv preprint arXiv:2201.03522, 2022a.
>
> [2] Yuling Yan, Gen Li, Yuxin Chen, and Jianqing Fan. Model-based reinforcement learning is minimax-optimal for offline zero-sum markov games. arXiv preprint arXiv:2206.04044, 2022.
>
> [3] Han Zhong,Wei Xiong, Jiyuan Tan, LiweiWang, Tong Zhang, ZhaoranWang, and Zhuoran Yang. Pessimistic minimax value iteration: Provably efficient equilibrium learning from offline datasets. arXiv preprint arXiv:2202.07511, 2022.
>
> [4] Wei Xiong, Han Zhong, Chengshuai Shi, Cong Shen, Liwei Wang, and Tong Zhang. Nearly minimax optimal offline reinforcement learning with linear function approximation: Single-agent mdp and markov game. arXiv preprint arXiv:2205.15512, 2022.
>
> [5] Qiwen Cui and Simon S Du. Provably efficient offline multi-agent reinforcement learning via
> strategy-wise bonus. Advances in Neural Information Processing Systems, 35:11739–11751, 2022.
>
> [6] Yuheng Zhang, Yu Bai, and Nan Jiang. Offline learning in markov games with general function approximation. In International Conference on Machine Learning, pp. 40804–40829. PMLR, 2023b.
>
> [7] Peter Sunehag, Guy Lever, Audrunas Gruslys, Wojciech Marian Czarnecki, Vinícius Flores Zambaldi, Max Jaderberg, Marc Lanctot, Nicolas Sonnerat, Joel Z Leibo, Karl Tuyls, et al. Value-decomposition networks for cooperative multi-agent learning based on team reward. In AAMAS, pages 2085–2087, 2018.
>
> [8] Tabish Rashid, Mikayel Samvelyan, Christian SchroederWitt, Gregory Farquhar, Jakob Foerster, and Shimon Whiteson. Qmix: Monotonic value function factorisation for deep multi-agent reinforcement learning. In International Conference on Machine Learning, pages 4292–4301, 2018.
>
> [9] Peng, B., Rashid, T., Schroeder de Witt, C., Kamienny, P. A., Torr, P., Böhmer, W., & Whiteson, S. (2021). Facmac: Factored multi-agent centralised policy gradients. Advances in Neural Information Processing Systems, 34, 12208-12221.

---

> ### Author Response · Authors · 2024-11-24
> **Additional Questions?**
>
> Dear reviewer,
>
> We have uploaded a revised version of our paper where we added discussion on the above related works. Since the discussion period is ending soon, we would like to kindly inquire whether our response has addressed your concerns. If you have additional questions, please do not hesitate to let us know. Thank you once again for your valuable feedback.

---

### Official Review · Reviewer_nwcs · 2024-11-04

**Soundness:** 2
**Presentation:** 3
**Contribution:** 2
**Rating:** 8
**Confidence:** 4

**Summary:**

This paper proposes a new notion of interaction rank to characterize the structure of a Markov game. If the true reward function and the learned reward function has a low interaction rank, then the authors designed decentralized \chi^2 regularized policy gradient approach and prove it converges to a CCE. The idea of low interaction rank is interesting. It may have potential in practice to help understand a wide range of multi-agent RL problems.

**Strengths:**

The notion of interaction rank is new and interesting. The authors provide statistical convergence and sample complexity guarantee under the assumption of low interaction rank for decentralized Markov game.

**Weaknesses:**

1. The assumption MGs with decoupled transitions is very restrictive.

2. Assumption that the reward function and transition kernel belong to the function class is restrictive. Further assumption that the function classes are finite are not practical at all.

3. The algorithm 1, line 3, estimating the transition kernel may not be as easy as it looks to be. Solving for the arg max in practice will be difficult.

4. In the discussion on page 9, the authors claim that existing studies are not decentralized. However, it looks to this reviewer that the advantage in this paper may come from the assumption of decoupled transition.

**Questions:**

1. Can the authors explain the on-support and off-support components in Theorem 2? The off-support component depends on the gap between the best response policy of agent i over the entire probability simplex and the best response policy within a bounded $\chi^2$ divergence ball. This seems to be due to the $\chi^2$ divergence regularizer in the algorithm. It is not clear here whether such a $\chi^2$ divergence regularizer is a wise choice, though it may provide some convenience for the analysis, it limits the performance of the obtained policy.

2. The authors then define the C_sin, which can be infinity due to taking the max over $\mu_i$ if $\nu_i$ is zero for some action a_i and context c. This reviewer is curious whether the single policy concentration coefficient can be defined here, and the optimality gap can be derived as a function of the single policy concentration coefficient. (similar comment for assumption 4).

3. Can the authors provide some examples where the low interaction rank assumption can be satisfied? If given unknown environment, how to verify such a low interaction rank assumption?

---

> ### Author Response · Authors · 2024-11-15
> **Response (Part 1)**
>
> Thank you for your review! Here are our response:
>
> 1. **The assumption of decoupled transition is restrictive**
>
> We first would like to clarify that there are no decentralized statistically and computationally efficient offline algorithms for the MARL setting with general function classes. This work can be seen as a first attempt, and there may be less restrictive assumptions that can be made. However, for the general case, there may not be efficient decentralized algorithms.
>
> Second, the decoupled transition finds many applications in practice. For example, in sensor networks, it is common that the state of each sensor (like its remaining battery) only depends on its past power consumption.  In many robotic scenarios, the location of the current robot also only depends on its own previous location and action.  Existing works have also investigated this special class of Markov games [1-3].
>
> On the other hand, our analysis can be naturally extended to Markov games with coupled transitions by leveraging the notion of ‘dependence level’ in [3]. Intuitively, the dependence level characterizes the error of utilizing a decoupled transition to approximate the original transition. We can show that our algorithm still works in this setting, with an additional approximation error term in the maximal gap. Note that in many cases this approximation error is small. For example, in autonomous driving, when the traffic is not very crowded, the state of each car only has a small dependence on other cars’ states and actions.
>
> That said, we agree that it would be interesting to extend our discussion to general MGs without incurring the approximation error. However, this is not trivial. To our knowledge, even in the full-information setting, there is no decentralized no-regret algorithm for general MGs with function approximation. Therefore, we would like to leave this generalization to future works.
>
> 2.  **The assumptions of function classes are restrictive**
>
> Realizability: Assuming the ground truth reward and transition belongs to the function classes is common in theoretical analysis [4-7]. That said, our analysis can be easily extended to the setting where the function classes have approximation error, which will only incur an additional approximation error term in the maximal gap.
> Finiteness:  We assume the function classes are finite to simplify our presentation, as also common in previous works that study RL in the function approximation setting. As discussed in the paper, our analysis and results still hold for infinite function classes and we only need to replace the cardinality of the function classes with its covering or bracketing number [8].
>
> 3. **Estimating the transition kernel is not easy**
>
> We use MLE to estimate the transition kernel, which is a common oracle in the existing literature [5-7]. In practice, this is indeed not difficult, as we can use the negative log-likelihood (NLL) loss function and gradient-based methods to implement MLE.
>
> We also would like to point out that the existing algorithms for provable offline general-sum MARL are much more complicated than MLE, which requires constructing a confidence set for every candidate joint policy and solving a maximin problem [9].
>
> 4. **Decentralization comes from decoupled transitions**
>
> We would like to clarify that this is incorrect. Let us consider the general-sum multi-agent contextual bandits where there is no transition. When you specialize the existing provable offline general-sum MARL methods to this setting [9,10], they are still centralized and computationally inefficient.  Our analysis shows that the interaction rank of the reward model allows us to control the sample complexity of a decentralized algorithm for finding approximate equilibrium in MARL. Even under the assumption that the transition model is decoupled, this fact is surprising, since reward models with low interaction rank may still depend on all agents in a complex manner. Technically, the reason why interaction rank structure is useful for the offline setting is the fact that such models are more robust to distribution shift, as we established in Section 3.

---

> > ### Author Response · Authors · 2024-11-15
> > **Response (Part 2)**
> >
> > 5. **Why use $\chi^2$ divergence**
> >
> > You are correct that the off-support component quantifies the performance difference between the global optimal policy and the optimal policy in the covered policy class, which in this case is the $\chi^2$ divergence ball. The reason why we use $\chi^2$ divergence is that theoretically KL divergence is not strong enough to keep the learned policy close to the behavior policy (as shown in a recent paper [11]) while $\chi^2$ divergence can.
> >
> > Meanwhile, we can choose a proper regularization coefficient $\lambda$ such that this regularization term will not influence the performance of the learned policy too much. As shown in Theorem 3, if the offline dataset has sufficient coverage, we are still able to learn a near-optimal policy and thus this $\chi^2$ divergence will not limit the performance of the learned policy.
> >
> > 6. **$C_{sin}$**
> >
> > $C_{sin}$ can be infinite. However, we introduce $C_{sin}$ only to compare our results with the existing work [10]. Our theorems (see Theorem 2 and Theorem 3) and analysis do not depend on this concentrability coefficient and still hold even when $C_{sin}$ is infinite, which is an advantage of our algorithm compared to [10]. This is because our algorithm uses regularization to keep the learned policy close to the behavior policy.
> >
> >
> >
> > 7. **When low interaction-rank is true**
> >
> > Low interaction-rank structures hold naturally in many existing systems. For example, the well-known polymatrix games [12-14], whose investigation started in the 1970s, characterize the reward function via pairwise interactions and thus its IR is 2. Another common example is the network games [2,15,16] where the reward only depends on
> > the neighbors and thus its IR is the degree of the network, which is often much smaller than the number of vertices in the network.
> >
> > Last but not least, we can approximate a general reward function with its $K$-order Taylor expansion, which will give us a $(K+1)$-IR game. In many cases, we only need a small $K$ to make the Taylor expansion close to the original reward. Therefore, when the exact IR is unknown, we can try critics with different IR in a small range (e.g., 2~10) and use the one with the best validation performance.

---

> > > ### Author Response · Authors · 2024-11-15
> > > **References**
> > >
> > > [1] Runyu Zhang, Yuyang Zhang, Rohit Konda, Bryce Ferguson, Jason Marden, and Na Li. Markov games with decoupled dynamics: Price of anarchy and sample complexity. In 2023 62nd IEEE Conference on Decision and Control (CDC), pp. 8100–8107. IEEE, 2023a.
> > >
> > > [2] Alex DeWeese and Guannan Qu. Locally interdependent multi-agent mdp: Theoretical framework for decentralized agents with dynamic dependencies. arXiv preprint arXiv:2406.06823, 2024.
> > >
> > > [3] Ruiyang Jin, Zaiwei Chen, Yiheng Lin, Jie Song, and Adam Wierman. Approximate global convergence of independent learning in multi-agent systems. arXiv preprint arXiv:2405.19811, 2024.
> > >
> > > [4] Tengyang Xie, Ching-An Cheng, Nan Jiang, Paul Mineiro, and Alekh Agarwal. Bellman-consistent pessimism for offline reinforcement learning. arXiv preprint arXiv:2106.06926, 2021.
> > >
> > > [5] Christoph Dann, Nan Jiang, Akshay Krishnamurthy, Alekh Agarwal, John Langford, and Robert E Schapire. On oracle-efficient pac rl with rich observations. Advances in Neural Information Processing Systems, 2018:1422–1432, 2018.
> > >
> > > [6] Alekh Agarwal, Sham Kakade, Akshay Krishnamurthy, and Wen Sun. Flambe: Structural complexity and representation learning of low rank mdps. arXiv preprint arXiv:2006.10814, 2020.
> > >
> > > [7] Masatoshi Uehara, Xuezhou Zhang, and Wen Sun. Representation learning for online and offline rl in low-rank mdps. arXiv preprint arXiv:2110.04652, 2021.
> > >
> > > [8] M.J. Wainwright. High-Dimensional Statistics: A Non-Asymptotic Viewpoint, volume 48 of Cambridge Series in Statistical and Probabilistic Mathematics. Cambridge University Press, 2019. ISBN 9781108498029.
> > >
> > > [9] Yuheng Zhang, Yu Bai, and Nan Jiang. Offline learning in markov games with general function approximation. In International Conference on Machine Learning, pp. 40804–40829. PMLR, 2023b.
> > >
> > > [10] Qiwen Cui and Simon S Du. Provably efficient offline multi-agent reinforcement learning via
> > > strategy-wise bonus. Advances in Neural Information Processing Systems, 35:11739–11751, 2022.
> > >
> > > [11] Huang, A., Zhan, W., Xie, T., Lee, J. D., Sun, W., Krishnamurthy, A., & Foster, D. J. (2024). Correcting the mythos of kl-regularization: Direct alignment without overparameterization via chi-squared preference optimization. arXiv preprint arXiv:2407.13399.
> > >
> > > [12] Joseph T Howson Jr. Equilibria of polymatrix games. Management Science, 18(5-part-1):312–318, 1972.
> > >
> > > [13] Fivos Kalogiannis and Ioannis Panageas. Zero-sum polymatrix markov games: Equilibrium collapse and efficient computation of nash equilibria. Advances in Neural Information Processing Systems, 36, 2024.
> > >
> > > [14] Revan MacQueen and James Wright. Guarantees for self-play in multiplayer games via polymatrix decomposability. Advances in Neural Information Processing Systems, 36, 2024.
> > >
> > > [15] Andrea Galeotti, Sanjeev Goyal, Matthew O Jackson, Fernando Vega-Redondo, and Leeat Yariv. Network games. The review of economic studies, 77(1):218–244, 2010.
> > >
> > > [16] Chanwoo Park, Kaiqing Zhang, and Asuman Ozdaglar. Multi-player zero-sum markov games with networked separable interactions. Advances in Neural Information Processing Systems, 36, 2024.

---

> ### Author Response · Authors · 2024-11-24
> **Additional Questions?**
>
> Dear reviewer,
>
> We have uploaded a revised version of our paper. Since the discussion period is ending soon, we would like to kindly inquire whether our response has addressed your concerns. If you have additional questions, please do not hesitate to let us know. Thank you once again for your valuable feedback.

---

> > ### Comment · Reviewer_nwcs · 2024-11-24
> > **Thank you**
> >
> > I appreciate the authors efforts in addressing my concerns. I increased my score to 8 to reflect this.

---

> > > ### Author Response · Authors · 2024-11-24
> > >
> > > Thank you very much!

---

### Author Response · Authors · 2024-11-24
**Revised Version**

Dear reviewers,

We have uploaded the revised version of out paper. The major modifications are as follows.

1. We added additional related works on provable offline and decentralized MGs and factored critics in Appendix A to complement the background.

2. We moved MGs with decoupled transitions to preliminaries.

3. We clarified that we addressed both cooperative and competitive MARL in the abstract.

Thank you again for your feedback!

---

### Meta-Review · Area_Chair_17Xg · 2024-12-21

**Metareview:**

All reviewers appreciate the novelty of interaction rank, and its tight statistical characterization in multi-agent RL. Accept.

**Additional Comments On Reviewer Discussion:**

NA

---

### Decision · Program_Chairs · 2025-01-22

Accept (Poster)